

# Investigating the impact of sub-ice shelf melt on Antarctica Ice Sheet spin-up and projections

Fan Gao[1,2], Qiang Shen[1,2], Hansheng Wang[1,2], Tong Zhang[3], Liming Jiang[1,2], Yan Liu[4], C.K. Shum[5], Yan An[1,2], and Xu Zhang[1,2]

[1]State Key Laboratory of Precision Geodesy, Innovation Academy for Precision Measurement Science and Technology, Chinese Academy of Sciences, Wuhan 430077, China

[2]College of Earth and Planetary Sciences, University of Chinese Academy of Sciences, Beijing 100049, China

[3]State Key Laboratory of Earth Surface Processes and Disaster Risk Reduction, Faculty of Geographical Science, Beijing Normal University, Beijing 100875, China

[4]State Key Laboratory of Remote Sensing Science, College of Global Change and Earth System Science, Beijing Normal University, Beijing 100875, China

[5]Division of Geodetic Science, School of Earth Sciences, The Ohio State University, Columbus, Ohio, 43210, USA

*Correspondence to*: Qiang Shen (cl980606@whigg.ac.cn) and Tong Zhang (tongzhangice@gmail.com)

**Abstract.** Sub-ice shelf melting is critical for the stability of the Antarctic Ice Sheet, as it influences ice shelf
buttressing that impedes grounded ice flow. Previous studies have emphasized that uncertainties in the state of sub-ice shelf melting contribute to inaccuracies in future sea-level projections. To better understand how sub-ice shelf melt rates affect model initialization and predictions, we adopt a single ice sheet model (PISM) and investigate two different sub-ice shelf melt rate schemes during model spin-ups. We then drive the Antarctic Ice Sheet into the future using identical environmental forcings. We find that, despite closely matched steady-state geometries achieved through the
spin-up process with different sub-ice shelf melt rates, the prognostic simulations reveal significantly divergent ice mass changes, particularly in marine ice sheet regions. By 2100, the difference in global sea-level contributions from the Antarctic Ice Sheet can be as large as ~57%, primarily from West Antarctica. This discrepancy arises because the spin-up initialization method alters the ice sheet's dynamic state, such as basal friction and thermal regimes, leading to varied ice sheet mass changes. Therefore, this study underscores the importance of sub-ice shelf melting and ice
sheet model initialization methods in reducing uncertainties in predicting the Antarctic Ice Sheet's future.

## 1 Introduction

A substantial majority of Antarctica's grounded ice discharge through its fringing ice shelves, which provide critical buttressing to upstream ice mainly through two primary mechanisms: lateral shear stresses along sidewalls and basal resistance forces at pinning points on topographic highs (Schoof, 2007; Goldberg et al., 2009; Feldmann & Levermann,
2023; Feldmann et al., 2024; Miles & Bingham, 2024). The exposure of ice shelves to warm seawater causes basal melting, combined with their near-flotation elevations, resulting in high susceptibility to oceanic forcing (Bindschadler



et al., 2013; Depoorter et al., 2013; Li et al., 2023). Observations reveal accelerating ocean-driven thinning of Antarctic ice shelves over recent decades (Paolo et al., 2015; Rignot et al., 2019), where enhanced basal melting reduces buttressing effects and promotes grounding line retreat, which collectively represents the primary driver of increased
ice discharge (Jacobs et al., 2011; Pritchard et al., 2012; Seroussi et al., 2014; Jourdain et al., 2020; Reese et al., 2020). Particularly on retrograde bed slopes, such retreat may trigger Marine Ice Sheet Instability (MISI), which has been observed to increase grounded ice velocities by 34% (Rignot et al., 2008) and may amplify ice loss by up to 0.6 m of sea level equivalent (SLE) this century (DeConto & Pollard, 2016; Schlemm et al., 2022).

Methods for ice sheet models to represent sub-ice shelf melting include linear thermal forcing (TF-linear)
parameterization (Martin et al., 2011; Lowry et al., 2021), ice-shelf cavity models developed from box or plume models (Lazeroms et al., 2018; PICO, Reese et al., 2018; PICOP, Pelle et al., 2019), empirical approximations (Cornford et al., 2015; Cornford et al., 2020), basin-averaged melt estimates (Seroussi et al., 2019), and spatially partitioned quadratic parameterization (ISMIP6 protocol, Jourdain et al., 2020). The Antarctica initialization model intercomparison project has demonstrated that ice sheet models exhibit significant divergence in response to variations
in initial basal melt conditions, accounting for 5 % - 125 % of total mass change in initialization experiments (Seroussi et al., 2019; Seroussi et al., 2020). This pronounced model spread underscores persistent challenges in accurately representing sub-ice-shelf oceanic processes during ice sheet model initialization (Pritchard et al., 2012; Alevropoulos-Borrill et al., 2020), and may propagate into projection uncertainties, particularly for ice dynamics influenced by oceanic forcing.

However, previous model intercomparison projects combined ice sheet models with varying numerical complexities and initialization methods (e.g., spin-up and data assimilation). This diversity makes it difficult to explicitly identify the physical mechanisms driving the large range of projection uncertainties. Zhang et al. (2024) addressed this limitation by adopting a single ice sheet model (Community Ice Sheet Model, CISM; Lipscomb et al., 2019; Berdahl et al., 2023) to investigate the impacts of geothermal heat flux and basal sliding conditions on Greenland Ice Sheet
initialization. Extending this approach and considering the crucial role of ice shelves in Antarctica, we propose conducting similar experiments for the Antarctic Ice Sheet using the identical ice sheet model and initialization method to assess the impacts of sub-ice-shelf melt rates. This focused investigation will address two key questions: (1) How do varying sub-ice-shelf melt rates impact the model initialization state? (2) How do these melt rates affect long-term Antarctic Ice Sheet projections?

Therefore, in this paper, we consider two different sub-ice shelf melt rate approaches in the ice sheet model (Parallel Ice Sheet Model, PISM) by first spinning-up and then projecting the Antarctica Ice Sheet (AIS). The structure of this paper is organized as follows: Section 2 details the methodological approach and experimental design for projections. Section 3 presents the key simulation results, while Sections 4 and 5 provide comprehensive results and discuss the implications for ice dynamics and sea level rise projections.



## 2 Model and Methods

We conduct ice sheet simulations using the Parallel Ice Sheet Model (PISM v.1.0.7) (Bueler et al., 2007; Martin et al., 2011; Winkelmann et al., 2011; Albrecht et al., 2020), an open-source, three-dimensional thermomechanical coupled model that integrates ice dynamics and thermodynamics. PISM employs a hybrid stress balance strategy (Martin et al., 2011; Winkelmann et al., 2011) by combining the Shallow Ice Approximation (SIA) for grounded ice

(Gudmundsson, 2003; Bueler et al., 2007; Pollard & DeConto, 2012) and the Shallow Shelf Approximation (SSA) for floating ice (Hindmarsh, 2006; Bueler & Brown, 2009; Pollard & DeConto, 2012).

To keep model consistency and for the convenience of results comparison, we follow the same initialization configuration and initial conditions as in LOW21 but using a different observational sub-ice shelf melt rate obtained from satellite altimetry, radar, and other datasets (Rignot et al., 2013; Fig. 2; Table 1). We utilize the BedMachine v.3

dataset (Morlighem et al., 2019) for initial topography, encompassing ice thickness and bedrock topography. Air temperature and precipitation inputs are derived from RACMO 2.3p2, averaged over 1979–2014 (van Wessem et al., 2018). Surface mass balance is calculated using a degree-day model (Ohmura, 2001; Calov & Greve, 2017), with near-surface temperature locally adjusted based on elevation changes using a correction factor of 0.008°C/m (Pittard et al., 2022). The spatial distribution of oceanic conditions in our study is presented in Fig. 1. Based on this data, the

sub-shelf ice temperature serves as the boundary condition for energy conservation, and the sub-ice shelf mass flux is represented by either an observational melt rate dataset (our approach, S1) or a parameterized method (LOW21, S2):

$$S_1 = \rho_i B, \tag{1}$$

where $\rho_i$ indicates the ice density, and B represents the sub-ice shelf melt rates. On the other hand, LOW21 employs an ocean model with Southern Ocean temperature and salinity (Fig. 2, Schmidtko et al., 2014) to calculate the sub-

shelf mass flux via a TF-linear parameterization (Martin et al., 2011), the main calculation equation is as follows:

$$S_2 = \rho_{sw} c_m \gamma_T F_{melt} (T_s - T_f)/(L_i \rho_i), \tag{2}$$

where $\rho_{sw}$ denotes the seawater density, $c_m$ represents the specific heat capacity of the ocean mixed layer, $L_i$ refers to the latent heat of phase change for ice, $\gamma_T$ represents the thermal exchange velocity between seawater and ice (assigned $\gamma_T = 10^{-4}$; Holland and Jenkins, 1999; Hellmer and Olbers, 1989), $F_{melt}$ is a model parameter (assigned

$F_{melt} = 5 \times 10^{-3}$, Beckmann and Goosse, 2003), $T_s$ is the temperature of ocean water (assigned $T_s$=271.45 K, Beckmann and Goosse, 2003), and $T_f$ denotes the temperature of seawater at depth $z_b$ beneath the ice shelf:

$$T_f = 273.15 + 0.0939 - 0.057S_0 + 7.64 \times 10^{-4} z_b, \tag{3}$$





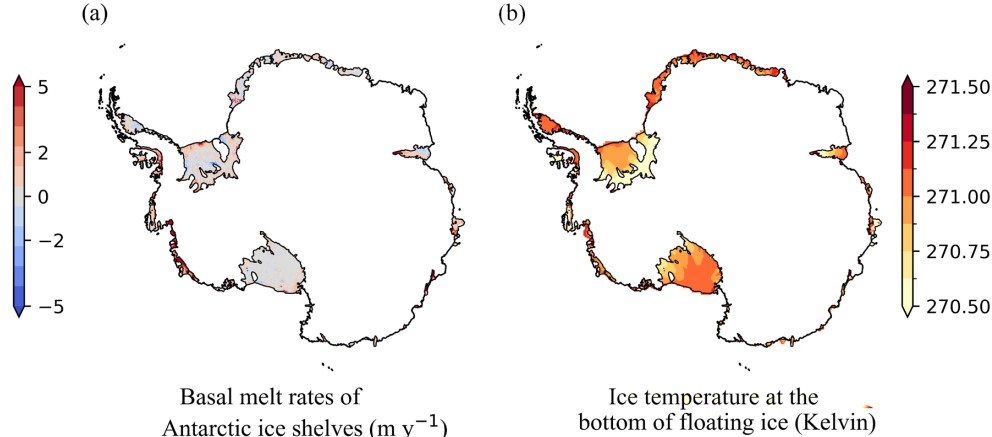

**Figure 1: Ocean conditions used in our simulations.** (a) observation of sub-ice shelf basal melt rates (Rignot et al., 2013); (b)

Temperature field beneath ice shelves (Chambers et al., 2021).

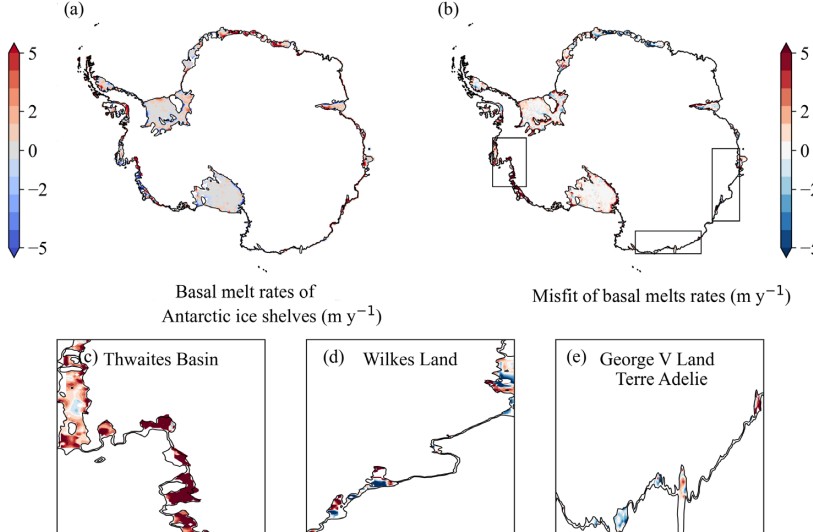

**Figure 2**: **Comparison of sub-ice shelf melt rates between our study and LOW21** (our study relative to LOW21). (a) LOW21

sub-ice shelf melt rates derived from ocean model. (b) Differences in basal melt rates between our study (Rignot et al., 2013) and

LOW21, with three black boxes highlighting the regions of interest: (c) Thwaites Basin, (d) Wilkes Land, and (e) George V Land

Terre Adelie.

During model initialization, a "multi-stage" spin-up procedure (Golledge et al., 2015; Lowry et al., 2021) is applied

to achieve a pseudo-equilibrium ice sheet state under constant climate conditions, with a 16 km spatial resolution : (1)

a brief 10-year smoothing utilizing the shallow ice approximation, (2) a 250,000-year simulation to allow the enthalpy

field to reach thermal equilibrium, (3) a 1,500-year evolutionary incorporating full model physics, including the

application of sub-ice shelf melt rates to constrain ice dynamics, and (4) a 65-year historical run to connect



initialization and prediction, during which the current ice sheet thickness is reconstructed. Further, we conduct projection experiments, initiated in 2015, by employing "high" and "low" scenarios controlled by the sub-grid melt interpolation, where "high" utilizes the scheme and "low" omits it (Albrecht et al., 2011; Golledge et al., 2015). These experiments used climate forcing derived from the CMIP5 IPSL-CM5A-MR (Barthel et al., 2020; Payne et al., 2021)

and the CMIP6 CNRM-CM6-1 (Nowicki et al., 2016; Kamworapan et al., 2021) to assess Antarctica's contribution to global mean sea level rise by 2100.

### 3 Model Initialization Results

#### 3.1 Comparison with the case of different sub-ice shelf melt rates

We compare our simulations with those of LOW21 in terms of ice thickness and surface velocity, using observational

datasets (BedMachine v.3; MEaSUREs Phase-Based Antarctica Ice Velocity Map v.1), with RMSE difference of 2 m for ice thickness and 3 m $y^{-1}$ for surface velocity. Figure 3 shows the differences between model results and observations: the left column displays discrepancies for our simulations, while the right column shows those for LOW21. The comparison indicates generally consistent mass distribution and ice flow dynamics. To better highlight the differences between our simulations and LOW21 in ice thickness and velocity over Thwaites Glacier, we selected

a transect (Fig. 7) where discrepancies were most pronounced. Along this transect, we compare the grounding line positions, ice thickness, and surface velocity profiles between our model and LOW21.

We also compared our simulations and LOW21 against the Antarctica ice sheet model initialization results (initMIP-Antarctica) that employed the PISM (Seroussi et al., 2019). Both studies align with ensemble trends in ice mass, ice sheet area, ice shelf area, and potential sea level contributions. Our simulations exhibit minor differences in total ice

sheet mass (-6% to +11%) and ice area (-7% to -1%) compared to initMIP-Antarctica. Notably, deviations in potential sea level contributions are more pronounced (-17% to -2%), while ice shelf area discrepancies reach 44% relative to the DMI_PISM simulation (Fig. 4).

Overall, while we see minor differences between our results and that of the initMIP-Antarctica ensemble simulations, both our spinned-up ice volume (25.81×$10^6$ km$^3$) and the LOW21 experiment (25.77×$10^6$ km$^3$) exhibit close agreement

with the observed total ice volume (BedMachine v.3; 26(±0.4) ×$10^6$ km$^3$). This validates the robustness of our initialization configuration and lends us confidence for future projection experiments.



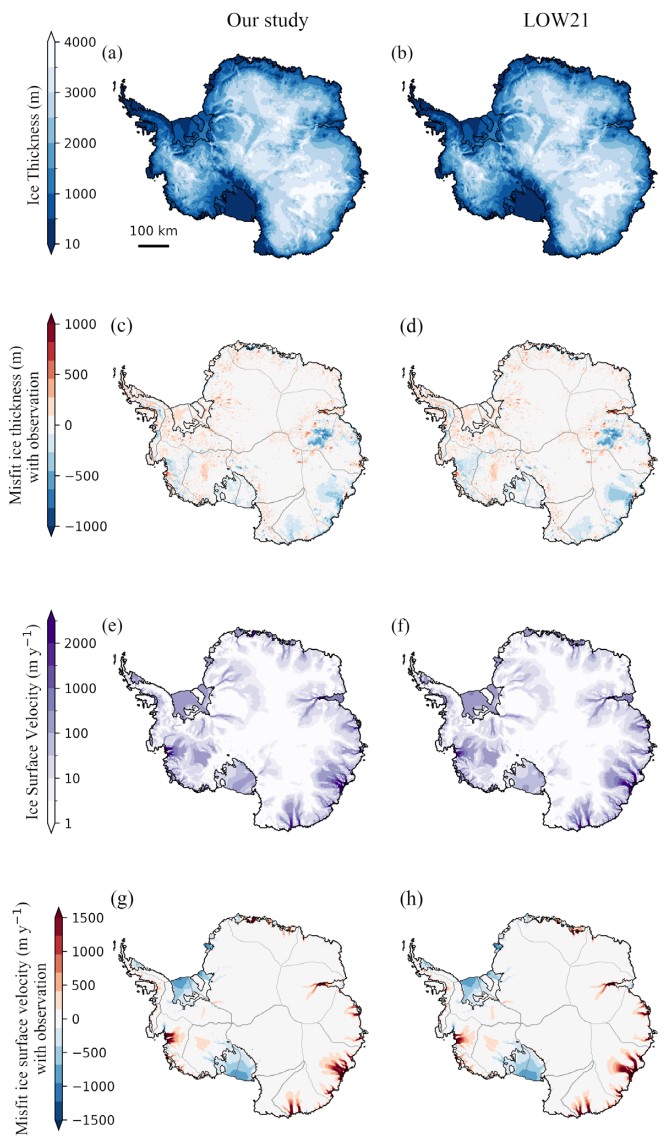

**Figure 3: Comparing our simulation (left column) and LOW21 (right column) relative to observations** (Morlighem et al., 2019; Mouginot et al., 2019).



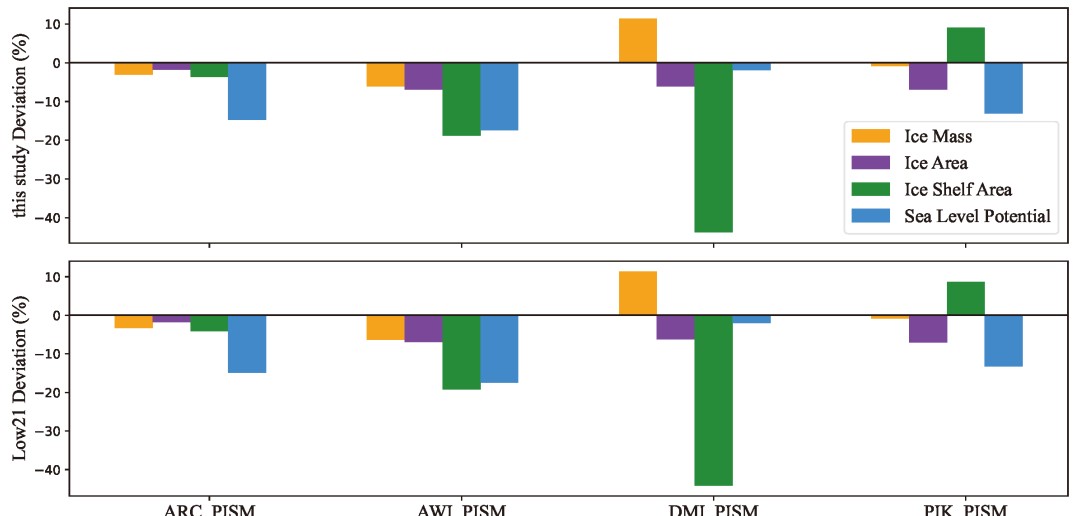

**Figure 4: Percentage deviations in steady-state metrics (ice mass, ice sheet area, ice shelf area, potential sea level contribution) relative to the initMIP-Antarctica PISM-based ensemble** (Seroussi et al., 2019). The vertical axis represents the percentage deviation between the results of our study, LOW21, and the simulations from participating institutions (*_PISM, where * denotes the institution abbreviation).

## 3.2 Differences in Marine Ice Sheet Regions

As clearly can be seen in Fig. 5, there are significant velocity differences in three marine-based regions characterized by retrograde bed slopes: Thwaites Basin (TB) in West Antarctica (WAIS), Wilkes Land (WL), and George V Land–Terre Adelie (GVL) in East Antarctica (EAIS). These regions are particularly susceptible to MISI due to their subglacial topography (Joughin et al., 2014; Mengel & Levermann, 2014; Greenbaum et al., 2015).

In Thwaites Basin, the observed sub-ice shelf melt rates (17.7 m $y^{-1}$ beneath Thwaites ice shelf, Fig. 1) accelerate ice surface velocity downstream, with a corresponding 74 m $y^{-1}$ RMSE difference from LOW21 (Fig. 5f). This led to around 40 m more ice thinning near the grounding line and an approximately 30 km more grounding-line retreat (Fig. 7), compared to the case in LOW21, while most upstream areas exhibit positive thickness anomalies (mean 49.5 m), suggesting complex feedback in ice dynamics. The ice volume above flotation in TB in our study shows a 5.5-fold bias reduction (-0.59%, 1.19 m SLE) compared to LOW21 (-3.28%, 1.16 m SLE), aligning closely with observations (1.20 ± 0.02 m SLE, Table 1).

In Wilkes Land, the Totton Glacier exhibits increased ice surface velocity under observed melt rates, yielding a 44 m $y^{-1}$ lower RMSE in our study relative to LOW21 (Fig. 5g), leading to regional mean ice thinning of 38.5 m. This generates stronger lateral resistance for adjacent glaciers, subsequently reducing ice flow in glaciers upstream of the Voyeykov and Moscow Ice Shelves, exhibiting a mean thickness anomaly of +39.2 m across these regions. The ice volume bias in WL decreased to -4.61% (6.63 m SLE) in our results, compared to -5.14% (6.59 m SLE) in LOW21, achieving a 10% improvement relative to the observed 6.95 ± 0.09 m SLE (Table 1).





In George V Land–Terre Adelie, the enhanced flow of the Ninnis Ice Shelf results in increased ice discharge and regional mean thinning (33.7 m). Conversely, the Cook and Mertz Ice Shelves and their upstream glaciers experience

reduced ice flux, causing regional ice thickness to increase by 25.5 m on average relative to LOW21. Our simulations demonstrate a reduced ice volume bias of -5.23% (3.35 m SLE) in WL, outperforming LOW21's -5.42% (3.34 m SLE) and reflecting closer agreement with the observed 3.53 ± 0.04 m SLE (Table 1).

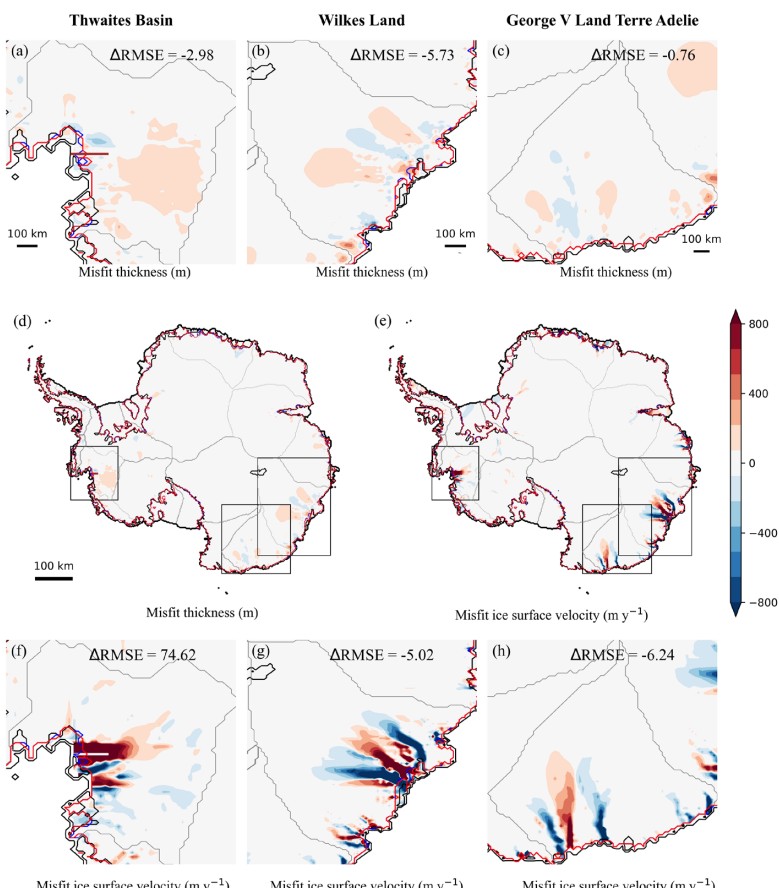

**Figure 5: Comparison of spinned-up ice thickness and velocity misfits between our results and LOW21** (our study relative

to LOW21). (a–c) Ice thickness differences in the TB, WL, and GVL, respectively. (f–h) Ice surface velocity misfits in the TB, WL, and GVL, respectively. The difference in root mean square error (ΔRMSE) between our results and LOW21, compared to observations. (d) and (e) present the deviations in ice thickness and surface velocity (our study relative to LOW21) between the two simulations, with three black boxes highlighting regions showing the most significant discrepancies. Grounding lines: LOW21 (blue), our study (red), and observed data (black) from BedMachine v.3 (Morlighem et al., 2019). The profile line locations

corresponding to Fig. 7 are in Thwaites Glacier: (a), (d) brown; (f) white.

A comparison of simulation results between our study and LOW21 for three marine ice sheet basins (TB, WL, and GVL; Table 1) reveals that the ice sheet model driven by observed sub-ice shelf melt rates achieves slightly better





alignment with observations. Although the RMSE of ice surface velocity increases (74 m y$^{-1}$ in the TB compared to LOW21 (Fig. 5f), the overall ice volume bias decreases by 3%, while volume biases for WL and GVL reduced by

0.5% and 0.2% (Table 1), respectively.

**Table 1:** Ice volume above flotation (m SLE) in three marine ice sheet basins, simulated at 16 km resolution.

| Basins | Observation | Our study | Misfit | LOW21 | Misfit |
|---|---|---|---|---|---|
| Thwaites Basin (TB) | 1.20 (±0.02) | 1.19 | -0.59% | 1.16 | -3.28% |
| Wilkes Land (WL) | 6.95 (±0.09) | 6.63 | -4.61% | 6.59 | -5.14% |
| George V Land Terre Adelie (GVL) | 3.53 (±0.04) | 3.35 | -5.23% | 3.34 | -5.42% |

### 3.3 Marine Ice Sheet dynamics during model spin-up

In WAIS, particularly for glaciers adjacent to the Amundsen Sea Embayment, the subglacial bedrock topography lying below sea level amplifies the sensitivity to ocean-driven forcings (Pritchard et al., 2012). Previous studies stated that

the Aurora Subglacial Basin in WL and the Wilkes Subglacial Basin in GVL, in EAIS, are characterized by extensive sedimentary basins that are highly susceptible to warming ocean conditions (Aitken et al., 2014; Frederick et al., 2016; Noble et al., 2020). These basins are also subject to an active subglacial hydrology process (Wright et al., 2012), and evidence of ocean-driven dynamic ice loss has been documented along the ice sheet margin (Li et al., 2016). The interplay between oceanic forcing, subglacial hydrology, and sedimentary geology significantly influences ice sheet

dynamics in these regions.

In this study, the application of observed basal melt rates with enhanced oceanic forcing (Fig. 2) intensifies ice-shelf basal melting, leading to geometric thinning and reduced buttressing effect of upstream ice flow (Gudmundsson, 2013; Miles et al., 2022). This triggers grounding line retreat, accelerating ice velocity. Accelerated ice flow substantially amplifies strain rates, leading to enhanced dissipative heating (Cuffey & Paterson, 2010; Dawson et al., 2022), and

therefore increasing temperatures at the basal ice layer (Fig. 6a-c). It then promotes basal melting (Fig. 6d-f) while reducing ice viscosity via thermal softening, collectively facilitating enhanced deformation and potentially increasing ice sheet destabilization (Hindmarsh, 2006; Adams et al., 2021). Additionally, subglacial meltwater lubricates the ice-bed interface, reducing basal friction through decreased effective pressure and accelerating ice flow (Fig. 6g-l). Enhanced sliding generates additional strain heating (Garbe et al., 2020), which promotes further basal melting and

meltwater production. This positive feedback process—where elevated basal water consent persistently reduces resistance to ice motion (Zhao et al., 2025)—is termed the basal thermal-hydrological feedback (Fowler et al., 2001; Clarke, 2005; van Pelt & Oleremans, 2012), facilitating the speed of ice sliding and ultimately leading to ice thinning.

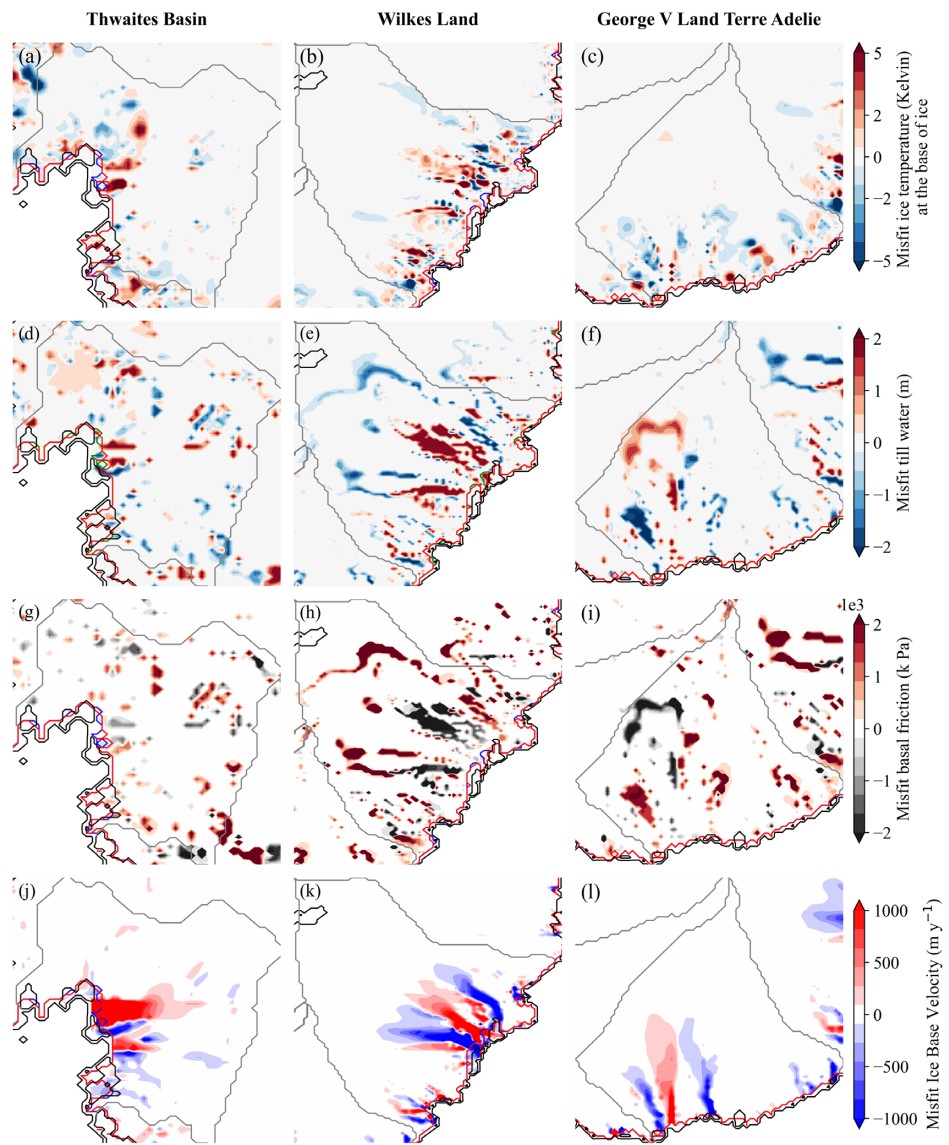

**Figure 6: Spatial distribution of misfits (our study relative to LOW21) in basal ice temperature field (Kelvin) (a-c), basal till water content (m) (d-f), basal friction (k Pa) (g-i), and basal velocity (m y⁻¹) (j-l) in the three basins.** The blue and red lines indicate the grounding line positions for LOW21 and our study, respectively, while the black line represents the observed grounding line from BedMachine v.3.

## 3.4 Grounding line location comparison

The impact of sub-ice shelf melt rates on ice sheet initialization can also be seen at the locations of the grounding line. Particularly, in WAIS, the retrograde bed topography amplifies susceptibility to MISI (Pritchard et al., 2012; Ritz et



al., 2015), rendering it highly responsive to ocean forcing. The observed sub-ice shelf melt rates applied in our simulations exceed LOW21's parameterized values by roughly 5 m yr[-1] beneath Thwaites and Pine Island ice shelves (Fig. 2). During spin-up, these elevated basal melt rates trigger MISI more easily, causing the grounding line on retrograde bedrock to retreat continuously until reaching a new steady state (Rignot et al., 2019; Li et al., 2022). Cross-

sectional analysis of Thwaites Glacier (Fig. 7) demonstrates this mechanism, with enhanced basal melting, causing an approximately 30 km grounding line retreat from its stabilized position (LOW21, purple line in Fig. 7) to a new quasi-stable state (our study, orange line in Fig. 7). This retreat increases ice discharge due to the reduced ice shelf buttressing effect, resulting in roughly 40 m ice thinning proximal to the grounding line and an anomalous nearly twofold acceleration in ice surface velocity compared to the LOW21 results. This feedback highlights how ocean-forced basal

melting propagates through ice sheet dynamics processes to alter initial ice geometry.

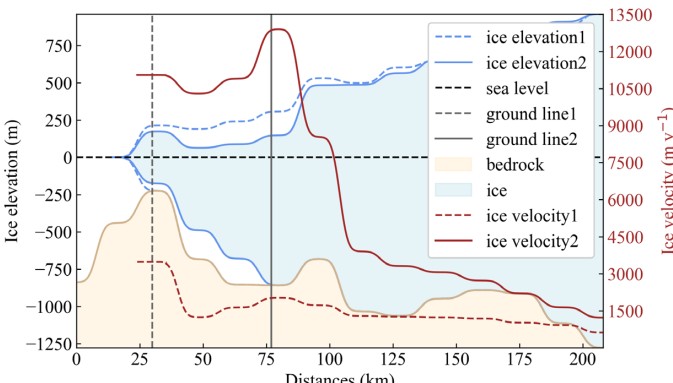

**Figure 7: Comparison between our simulation and LOW21 along the Thwaites Basin transect.** Ice elevation (blue line), ice surface velocity (brown line), and grounding line position (grey line) are distinguished by 1 and 2, based on LOW21 and our study, respectively. Sea level: black dashed line.

In fact, the retreat of the grounding line varies across different ice streams, influenced by factors such as basin size, water depth, bed roughness, neighboring ice shelf basal melt rates, and ice velocities (Pittard et al., 2022). Consequently, the discrepancies between observed and simulated grounding line positions differ across various regions of the AIS. For instance, the grounding line of the Siple Coast on the west side of the Ross Ice Shelf in our simulation agrees with LOW21 but extends further to the nearshore compared to observation (Fig. 5d-e). This

discrepancy can be attributed to the reversibility of grounding line migration on a retrograde-slope bedrock, which is characterized by oscillatory shifts (Martin et al., 2011; Pattyn et al., 2012). Notably, under a consistent model parameter configuration, the inclusion of observed sub-ice shelf melt rates did not significantly alter the steady-state grounding line migration position across the whole AIS, except within three marine-based regions.

## 4 Model Projection Results



### 4.1 Global mean sea level contribution from AIS

Prognostic simulations from 2015 to 2100 revealed divergent ice mass changes compared to LOW21, particularly in WAIS. Under various climate scenarios, our model projected AIS contributions to sea level rise ranging from 0.20 to 0.52 m SLE (Fig. 8), compared to 0 to 0.32 m SLE in LOW21—an average increase of roughly 0.18 m SLE (~57%). This discrepancy is driven primarily by enhanced ice loss from West Antarctica (0.29–0.34 m SLE, Table 2), where high sub-ice shelf melt rates triggered MISI more readily, whereas East Antarctica and the Antarctic Peninsula (AP) show minimal sea level contributions by 2100, i.e., 0.01–0.02 m SLE and 0.0011–0.0045 m SLE, respectively (Table 2). However, the projections of Antarctic ice sheet contributions to sea-level rise from 2015 to 2075 exhibit no significant dependence on emission scenarios, with substantial overlap in prediction ranges (Fig. 8c), which is relative to the hysteretic response of ice sheet dynamics to climate forcing (Garbe et al., 2020).

The SSP scenarios simulate higher warming magnitudes (averaging +0.14-0.25 °C) than RCP scenarios at equivalent radiative forcing levels (Tokarska et al., 2020; Wyser et al., 2020; Rounce et al., 2023), driving substantial divergences in AIS projections. By 2100, AIS contributions to sea-level rise under SSP5-8.5 reach 0.36 m SLE—12.5% higher than the RCP8.5 equivalent (0.32 m SLE)—with RCP high-emission projections even matching SSP low-scenario results (Fig. 8d, Table 2). Based on 2100 trajectory extensions, persistent ice mass is projected through the early 22nd century under SSP scenarios, whereas CMIP5-based RCP simulations indicate stabilizing ice-mass trends beyond 2100. Consequently, under anthropogenic warming, Antarctic contributions to sea-level rise in SSP high-risk scenarios demand heightened scientific attention due to amplified ice-climate feedback.

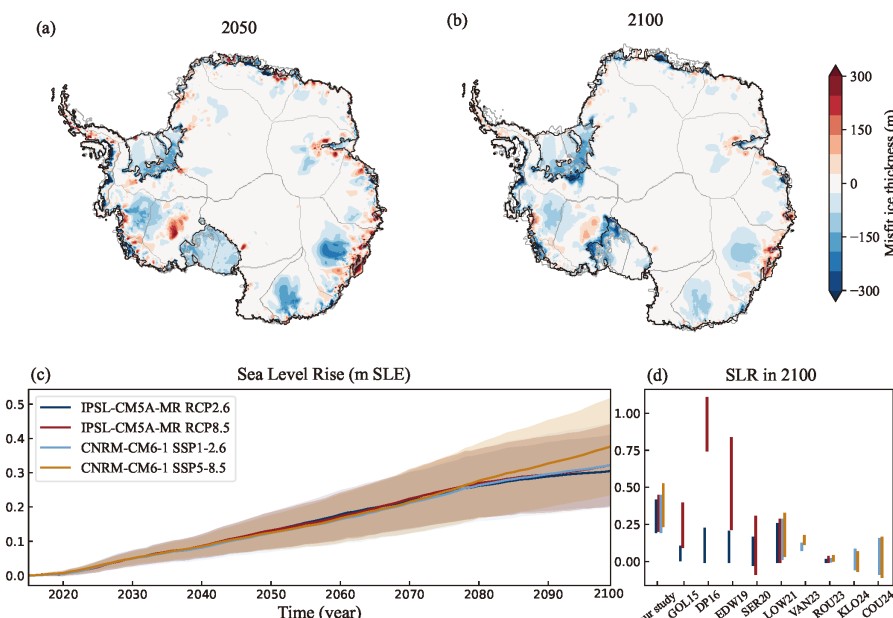

**Figure 8: Ice sheet thickness misfits and projected contribution of the AIS to sea level rise (m SLE).** Mean ice thickness spatial differences (our study relative to LOW21) in 2050 (a) and 2100 (b), under RCP scenarios. (c) Predicted sea level rise from





"high" and "low" simulations under four scenarios (color shading) and mean values (color lines). (d) Comparison of projected contributions by 2100 between our study and other studies: CHU13 (Church et al., 2013), GOL15 (Golledge et al., 2015), DP16 (DeConto & Pollard, 2016), EDW19 (Edwards et al., 2019), SER20 (Seroussi et al., 2020), LOW21 (Lowry et al., 2021), VAN23 (van der Linden et al., 2023), KLO24 (Klose et al., 2024), and COU24 (Coulon et al., 2024).

## 4.2 Regional Contributions to Global Mean Sea Level Rise

To explore the spatially variable response of the AIS, we partitioned the ice sheet into seven sectors based on their adjacency to surrounding oceans (Fig. 9). Following the IMBIE (Zwally et al., 2012), we further subdivided East Antarctica into the East Indian Ocean (EIO) and West Indian Ocean (WIO) sectors. This regional breakdown reveals stark contrasts in the mechanisms and magnitudes of sea level rise across the WAIS, EAIS, and AP (Fig. 9, Table 2), highlighting their distinct sensitivities to climate forcing. The WAIS emerges as the primary driver of AIS-related sea level rise, contributing 0.29–0.34 m SLE by 2100. This significant mass loss is propelled by anthropogenic greenhouse gas emissions, which have significantly altered shelf-break wind patterns over the Amundsen and Bellingshausen Seas (Holland et al., 2019; Noble et al., 2020). These altered winds facilitate greater intrusion of warm Circumpolar Deep Water onto the continental shelf, intensifying basal melting beneath ice shelves (Dinniman et al., 2016; Noble et al., 2020; Li et al., 2023). The resultant thinning diminishes ice shelf buttressing, accelerating the discharge of grounded ice, particularly from the Amundsen Sea (AS) and Bellingshausen Sea (BS) Embayment. Our simulations suggest that the mass loss of these sectors accounts for roughly 55% of WAIS's total contribution (Table 2), highlighting their critical influence on future sea level trends.

The EAIS presents a more complex picture, with a net contribution of 0.01–0.02 m SLE by 2100. While the integrated signal is small, it masks pronounced regional heterogeneity in mass changes. The West Indian Ocean (WIO) sector presents a net mass gain, consistent with observational trends (Boening et al., 2012) because enhanced moisture transport from the Southern Ocean drives increased surface accumulation. This marginal gain, however, is counteracted by ice dynamical adjustments within the East Indian Ocean (EIO) sector, specifically across the WL, where enhanced oceanic thermal forcing drives accelerated ice mass loss from the dynamically vulnerable Totten Glacier (Konrad et al., 2018). This contrast between surface mass balance gains and ice dynamic losses underscores the spatially heterogeneous response of EAIS, modulated by regional bathymetry and ocean-driven melt.

The AP plays a comparatively minor role in sea level rise, with contributions ranging from 0.0011–0.0043 m SLE by 2100. Peak mass loss occurs between 2075 and 2079, reaching 0.0061 m SLE under RCP 8.5 and 0.0045 m SLE under SSP 5-8.5, followed by a gradual decline (Fig. 9). This transient pattern is tied to the intensification of polar westerly winds, which partially offset warming-induced ice discharge by enhancing snowfall in the northern AP, thus creating a negative feedback mechanism that suppresses AIS mass loss (Goodwin et al., 2016). Given its limited ice volume, however, the AP's overall impact on sea level rise remains marginal. The findings underscore the divergent climate responses of EAIS, AP, and WAIS. While EAIS and AP exhibit mass gain or loss depending on the balance between accumulation and ablation, WAIS is primarily driven by dynamic mass loss resulting from changes in oceanographic conditions.





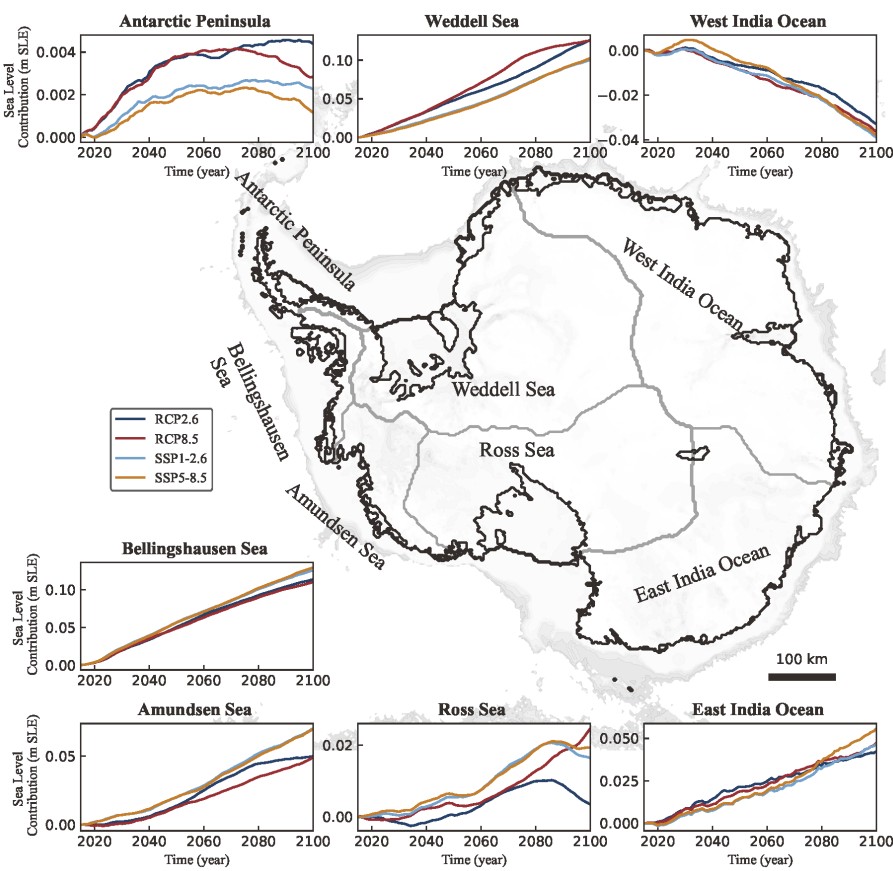

**Figure 9: The mean contribution of the seven sectors of AIS to sea level rise (m SLE) from 2015 to 2100.** Colored lines correspond to different scenarios.

**Table 2:** Sea level contribution (m SLE) of Antarctic Ice Sheet Basins by 2100.

| Region | RCP 2.6 | RCP 8.5 | SSP 1-2.6 | SSP 5-8.5 |
|---|---|---|---|---|
| Bellingshausen Sea (BS) | 0.1138 | 0.1102 | 0.1256 | 0.1297 |
| | (0.0857, 0.1419) | (0.0813, 0.1391) | (0.0982, 0.1530) | (0.1064, 0.1585) |
| Amundsen Sea (AS) | 0.0499 | 0.0493 | 0.0701 | 0.0700 |
| | (0.0405, 0.0593) | (0.0386, 0.0599) | (0.0553, 0.0850) | (0.0548, 0.0968) |
| Ross Sea (RS) | 0.0034 | 0.0247 | 0.0163 | 0.0193 |
| | (-0.0089, 0.0157) | (-0.0009, 0.0503) | (-0.0107, 0.0435) | (-0.0048, 0.0552) |
| Weddell Sea (WS) | 0.1260 | 0.1249 | 0.1005 | 0.1025 |
| | (0.0759, 0.1762) | (0.0734, 0.1764) | (0.0542, 0.1467) | (0.0582, 0.1638) |
| West Antarctica Ice Sheet | 0.2931 | 0.3090 | 0.3126 | 0.3444 |
| (WAIS) | (0.1932, 0.3931) | (0.1924, 0.4257) | (0.197, 0.4282) | (0.2146, 0.4743) |





| | | | | |
|---|---|---|---|---|
| West India Ocean (WIO) | -0.0330 (-0.0448, -0.0211) | -0.0363 (-0.0560, -0.0166) | -0.0389 (-0.0531, -0.0247) | -0.03.79 (-0.0471, -0.0227) |
| East India Ocean (EIO) | 0.0423 (0.0269, 0.0578) | 0.0473 (0.0277, 0.0669) | 0.0467 (0.0284, 0.0650) | 0.0559 (0.0467, 0.0867) |
| East Antarctica Ice Sheet (EAIS) | 0.0093 (0.0058, 0.0130) | 0.0110 (0.0109, 0.0111) | 0.0078 (0.0037, 0.0119) | 0.0180 (0.0240, 0.0396) |
| Antarctica Peninsula (AP) | 0.0043 (0.0024, 0.0062) | 0.0028 (0.0007, 0.0049) | 0.0022 (0.00004, 0.0044) | 0.0011 (-0.0021, 0.0034) |
| Antarctica Ice Sheet (AIS) | 0.3067 (0.2014, 0.4123) | 0.3228 (0.2042, 0.4415) | 0.3226 (0.2007, 0.4445) | 0.3635 (0.2365, 0.5173) |

### 4.3 Comparison with previous projections

Our projections of AIS contributions to sea level rise by 2100 under the SSP 5-8.5 scenario diverge significantly from previous estimates, particularly for WAIS. While ensemble projections from the Coupled Model Intercomparison Project Phase 6 models (CMIP6, Edwards et al., 2021) suggest that WAIS contributions range from -0.04 to 0.11 m SLE, our study predicts a significantly higher contribution of 0.20–0.47 m SLE. For the EAIS and AP, our projected sea level contributions (0.03–0.04 m SLE and -0.0021–0.0034 m SLE, respectively; Table 2) align closely with the emulator results (-0.05–0.06 m SLE and -0.01–0.02 m SLE, respectively), with WAIS emerging as the dominant divergence source.

Compared to ISMIP6 (Ice Sheet Model Intercomparison for CMIP6) Antarctic projections under RCP 8.5 (Seroussi et al., 2020), our WAIS contribution exceeds it by approximately 0.15 m SLE, with AP showing a slight increase (~0.002 m SLE) and EAIS exhibiting a minor reduction (~0.02 m SLE). These results demonstrate significant deviations in WAIS sea level contributions compared to prior projections, consistent with predicted ice thickness variations between our study and LOW21 (Fig. 8). The pronounced discrepancies in WAIS primarily stem from its vulnerability to oceanic forcing and complex bed topography, projecting an average ice thickness decrease of 50 m by 2100 relative to LOW21 (Fig. 8). While this study used parameterized melt rates, our incorporation of observational values reveals enhanced basal melting in critical WAIS regions like TB (Fig. 2), accelerating dynamic ice loss through processes such as MISI. In contrast, EAIS regions (WL and GVL) exhibit minor variations, contributing little to sea level rise during projection periods (2050 and 2100; Fig. 8). This regional contrast highlights WAIS's dominant role in creating divergence from earlier projections, demonstrating its heightened vulnerability to ocean-induced melt rates.

### 5 Model Uncertainties

We maintained consistent model configurations and climate forcings with LOW21 but constrained sub-ice shelf melt rates using observational data during spin-up, replacing their TF-linear parameterization. The elevated sub-ice shelf melt rates in our simulation (Fig. 2), notably in the Thwaites and Pine Island shelves, modify the initial AIS state.



Through an iterative spin-up process, the model iteratively adjusts key variables such as basal ice temperature field, basal friction coefficients, and grounding line positions to minimize discrepancies between simulation and observation. While the ice sheet geometries for LOW21 and our study are initialized with identical spin-up, distinct sub-ice shelf melt rates produce divergent ice thermodynamic states, causing the ice sheet to follow unique evolutionary trajectories in projection under identical external forcing, thus altering the potential contributions of sea level rise.

Compared to other prior studies, our sea level projections differ due to variations in ice sheet model configurations, including model resolution, ice dynamics (particularly stress balance schemes), represented physical processes (calving, hydrology, or bedrock uplift), and initialization methods (data assimilation or spin-up) (Seroussi et al., 2019; Levermann et al., 2020; Klose et al., 2024). Notably, the present-day AIS may not have been in a steady-state during the observational period, and thus, some of the misfits could be attributed to uncertainties in the observational data used for validation (Martin et al., 2011). Moreover, global climate models exhibit significant differences in projected global temperature increases, which in turn affect ice dynamics (Golledge et al., 2015; Schlegel et al., 2018; Klose et al., 2024). High-sensitivity climate models within the CMIP6 ensemble, such as IPSL-CM6A-LR (4.6°C), UKESM1-0-LL (5.3°C), and CESM2-WACCM (4.8°C), predict substantial warming over Antarctica, potentially driving extensive melting of the WAIS.

## 6 Conclusions

By initializing the ice sheet model with observed sub-ice shelf melt rates instead of the TF-linear parameterization employed in a previous study (LOW21), the ice geometry aligns closely with both observations and LOW21 after spin-up. However, our results reveal notable regional variations in ice sheet dynamics across three marine ice sheet regions compared to the parameterized method: the Thwaites Basin in West Antarctica, Wilkes Land, and George V Land–Terre Adelie in East Antarctica. In Thwaites Basin, elevated sub-ice shelf melt rates progressively trigger MISI, driving grounding line retreat that significantly weakens the ice shelf buttressing effect for upstream glaciers. Our simulations demonstrate 3 m ice thickness discrepancies and 74 m y$^{-1}$ ice velocity deviations compared to LOW21's simulations under observational validation. Variations in ocean forcing conditions in Wilkes Land and George V Land–Terre Adelie may alter the thermomechanical features at the grounded ice sheet, which then induce dynamic adjustments, causing approximately 6 m and 44 m y$^{-1}$ variations in ice thickness and surface velocity, respectively.

Despite using identical model configurations and future climate scenarios, our projections estimate a 57% higher contribution of sea level rise (~0.18 m SLE) by 2100 compared to the previous study (LOW21), due to a different treatment in prescribing sub-ice shelf melt rates during the PISM spin-up. The majority contributor to this SLE discrepancy stems from the Amundsen Sea sector in West Antarctica, a region typical of MISI, which also aligns with comparisons to other previous model projection results. In future modeling efforts, we suggest further efforts in investigating the sensitivity of Antarctica ice sheet model initializations to critical environmental factors before conducting fully prognostic Antarctica ice sheet simulations, to better constrain the projected ranges of global sea level rise.



**Code and Data Availability**

The Parallel Ice Sheet Model is freely available as open-source code from the PISM GitHub repository (https://github.com/pism/pism). Bedrock topography and ice thickness data are from the MEaSUREs BedMachine Antarctica, Version 3 compilation, available at https://nsidc.org/data/nsidc-0756/versions/3. Air temperature, precipitation, and geothermal heat flux inputs were taken from the ALBMAP version 1 compilation and can be downloaded from http://doi.pangaea.de/10.1594/PANGAEA.734145. Ice surface velocity used in validation may be obtained from MEaSUREs Phase-Based Antarctica Ice Velocity Map, Version 1, available at https://nsidc.org/data/nsidc-0754/versions/1.

**Author contribution**

Fan Gao, Qiang Shen, and Tong Zhang conceived and designed this experiment. Fan Gao performed data curation. Qiang Shen, Hansheng Wang, and Tong Zhang acquired funding. Qiang Shen provided resources. Fan Gao and Tong Zhang conducted the experiments. Fan Gao performed simulations. Liming Jiang and Yan Liu performed validation. C.K. Shum, Yan An, and Xu Zhang conducted visualization. Fan Gao wrote the original manuscript draft, and all authors contributed to reviewing and editing the manuscript.

**Competing interests**

There are no real or perceived conflicts of interest for any author.

**Acknowledgments**

We express our heartfelt gratitude to Prof. Lowry for his invaluable mentorship and significant contributions to this study. His provision of critical simulation datasets and expert guidance throughout the research process has been indispensable. We also sincerely thank Prof. Rignot for generously sharing observational data on Antarctic sub-ice shelf melt rates, which greatly enriched our work. Additionally, we extend our sincere appreciation to the Parallel Ice Sheet Model team for their continuous support and for developing and maintaining the PISM, which was essential to the success of this research.

**Financial support**

This work is supported by the National Natural Science Foundation of China (42374042, 42271133, and 42374045) and the Natural Science Foundation of Wuhan (2024040701010065).

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
