# Peer review of "Investigating the impact of sub-ice shelf melt on Antarctica Ice Sheet spin-up and projections"

_EGUsphere, 2025_

## Referee Comment (RC1)

**Overall Comments**

This manuscript addresses a scientifically important question about the sensitivity of Antarctic ice sheet projections to different sub-ice shelf melt rate treatments during model initialization. The core finding that the method of initialization of sub-ice shelf melt rates can lead to a significant difference in projected sea-level contributions is valuable for the ice sheet modeling community, and relevant to the scope of the journal.

**General Comments**

The research article presents novel insights by isolating the effects of sub-ice shelf melt rate treatment while maintaining identical model configurations, which addresses limitations of previous intercomparison studies that combined models of varying numerical complexities and initialization methods. The experimental design is well-motivated, and the writing is generally clear and well-structured.

The conclusions that West Antarctica dominates in driving projection uncertainties are potentially significant and well supported by the results, as are the substantial differences reported from previous studies. The comparison with CMIP6 and ISMIP6 projections raises important questions about model setup differences and other contributors beyond the melt rate parameterization that warrant further investigation. However, several methodological components require more detailed description to facilitate reproducibility, particularly the forward projection methodology, and the discussion of divergences from previous studies needs expansion.

**Specific Comments**

**Methodological Details:**

**Sub-grid interpolation:**

**p5, l107:** *"... we conduct projection experiments, initiated in 2015, by employing "high" and "low" scenarios controlled by the sub-grid melt interpolation"*

This sub-grid melt interpolation requires clearer explanation. What constitutes this interpolation scheme, and why does activating it constitute a "high" scenario versus omitting it for "low" scenarios? Clarify the distinction between these scenarios.

**Climate forcing implementation:**

**p5, l109-110:** " *These experiments used climate forcing derived from the CMIP5 IPSL-CM5A-MR (Barthel et al., 2020; Payne et al., 2021) and the CMIP6 CNRM-CM6-1 (Nowicki et al., 2016; Kamworapan et al., 2021) to assess …*"

I could also not find details on the implemented climate forcings, consider specifying:

- How are the forcing fields (temperature, salinity?) used/derived from these models?
- What is the temporal resolution of the forcing data?
- How are the CMIP5 or CMIP6 forcings interpolated or downscaled to the PISM grid? Describing this would be useful for reproducibility.

**Model configuration:**

**p11, l226:** "*consistent model configurations and climate forcings with LOW21..*"

It is not clear what consistent model configurations constitutes of, and the term is ambiguous. There is some clarification for this provided in Section 5, which would fit better in Section 2, where the experimental setup is first introduced. Consider establishing early on exactly which parameters remain identical between experiments and which differ.

**Forward projection:**

**P12, l231:** *"Prognostic simulations from 2015 to 2100 revealed divergent ice mass changes compared to LOW21, particularly in WAIS. Under various climate scenarios, …"*

The treatment of sub-ice shelf melt rates during the 2015-2100 projection period is unclear. Are the observational melt rates from Rignot et al. (2013) prescribed throughout the projections?

**Projection uncertainties:**

**p14, Table 2:**

The confidence intervals presented in Table 2 are not defined. Are these ranges derived from ensemble runs, sensitivity tests, or are a statistical treatment of the model output? Consider clarifying the source and methodology of these intervals and expand the table description/title accordingly.

**Initialization results:**

**Grounding line migration:**

**p11, l225:** *"...discrepancy can be attributed to the reversibility of grounding line migration on a retrograde-slope bedrock, which is characterized by oscillatory shifts"*

The differences presented among the three regions of retrograde bed slope (TB, WL, GVL) are interesting, and Section 3.3 provides a well written mechanistic understanding of the processes involved. In Section 3.4, the grounding line analysis mentions "reversibility of grounding line migration on retrograde-slope bedrock", raising an important point, but remains brief. Further, the phrasing of that sentence makes it unclear, as it seems to conflate the mechanism of oscillatory shifts in the grounding line with the bed geometry. A clearer explanation of the physical processes underlying these patterns in GL migration would be useful.

**Projection results:**

**p12, l238-239:** *"which is relative to the hysteretic response of ice sheet dynamics to climate forcing..."*

The lack of scenario dependence with significant overlap in the prediction ranges is consistent with the delayed/hysteretic response of ice sheet dynamics in that the current ice sheet state and near future (as during this 2015-2075 period) reflect historical forcing, but it is unclear from this sentence. Consider explaining/rephrasing.

**p12, l244-246:** *"...2100 trajectory extensions, persistent ice mass ....beyond 2100. "*

Are these extended projections beyond 2100 provided anywhere for this particular study, or does this statement refer existing literature? Consider expanding on this, particularly, which SSP scenario and which RCP simulations correspond to persistent ice mass and stabilizing trends.

**p15, l295-300:** *"... Compared to ISMIP6 (Ice Sheet Model Intercomparison for CMIP6) Antarctic projections under RCP 8.5 (Seroussi et al., 2020), our WAIS contribution exceeds it by approximately 0.15 m SLE, with AP showing a slight increase (~0.002 m SLE) and EAIS exhibiting a minor reduction (~0.02 m SLE..."*

The substantial differences from ISMIP6 projections raises important questions. To strengthen this section, please clarify whether the comparisons are made against the full ensemble or PISM-based runs alone. (e.g., from the description in LOW21, the model setup here would be most comparable to the VUW-PISM from ISMIP6). It would also be helpful to highlight and discuss any other model differences apart from the melt rate parameterization that could contribute to the divergences. If possible, a short comparison on the strengths/limitation of either this observationally constrained approach or the ISMIP6 approach would add significant value.

**p16, l318-321:** *"Compared to other prior studies, our sea level projections differ due to variations in ice sheet model configurations, including model resolution, ice dynamics (particularly stress balance schemes), represented physical processes (calving, hydrology, or bedrock uplift), and initialization methods (data assimilation or spin-up)"*

These lines address my above comment about detailing the potential reasons for the differences. Is there evidence to suggest which of these factors might be most important, and how it could be addressed in future work?

**Technical Comments**

**p1, l30-31:** This sentence on ice shelf susceptibility could be streamlined for clarity.

**p3, l74:** "(Rignot et al., 2013; Fig. 2; Table 1)" Should this reference be "Fig. 1"?

**p3, l90:** What is the depth of the ocean water temperature, T_s? Is it taken close to the bottom?

**p3, l92:** Define S_o (ocean salinity?) in Equation (3)

**p8, l166-170:** In Fig. 5 caption, clearly define ΔRMSE (Is it RMSE (current vs. obs.) – RMSE (LOW21 vs. obs.)?)

**p9, l175:** The percentages mentioned (0.5%, 0.2%) appear to conflict with the values presented in Table 1. Clarify whether these refer to overall ice volume biases or volume above flotation biases.

**p9, l186:** "enhanced oceanic forcing" needs a clearer definition in this context.

**p9, l195:** "elevated basal water consent": Should be "...content"

**p11, l211-212:** Line colors mentioned here are inconsistent with Fig. 7 description. (purple/orange in text vs. grey (dashed/solid) in figure)

**p12, l243:** "persistent ice mass": Should this be "ice mass loss"?

**p12, l243-247:** There seems to be a causal gap between the two sentences. Consider adding a brief description in between to establish the causal link between persistent ice mass (loss?) / stabilizing trends and "amplified ice-climate feedback" or modifying the sentence.

**p12, l250:** Fig. 8 caption should specify which RCP scenarios are used for panels (a) and (b).

**Secondary comments:**

The below comments are merely secondary suggestions for the authors, and I leave it to them whether to address these or not.

**Use of Rignot et al., (2013) melt rates:**

The reliance on Rignot et al., (2013) melt rate observations is understandable given their wide use in the modeling community. At the same time, this data now reflects conditions around a decade old, and more recent work, has further revealed interannual variability in ice shelf basal melt rates (Adusumilli et. al., 2020), and even slowdown in melt-driven thinning for certain sectors (Paolo et. al., 2023). I encourage the authors to briefly discuss and frame the use of this dataset within the context of more recent observational work.

**Validation against observations:**

**p16, l321:** *"...the present-day AIS may not have been in a steady-state during the observational period, and thus, some of the misfits could be attributed to uncertainties in the observational data used for validation"*

The authors note that the present-day AIS may not have been in steady state during the observational period. A brief discussion of the uncertainties and potential biases in the datasets used for validation would strengthen this argument.

Paolo et al., 2023, which constructed 3 km resolution datasets of ice thickness also revealed a **slowdown in thinning** from around 2008, specifically in the Amundsen, Bellingshausen and Wilkes sectors. Following from my earlier comment about melt rate observations, in addition to the citations in the introduction for accelerated thinning of ice shelves, this article may be worth including as a reference. The specific paper I'm referring to is:

Paolo, F. S., Gardner, A. S., Greene, C. A., Nilsson, J., Schodlok, M. P., Schlegel, N.-J., and Fricker, H. A.: Widespread slowdown in thinning rates of West Antarctic ice shelves, The Cryosphere, 17, 3409–3433, https://doi.org/10.5194/tc-17-3409-2023, 2023.

---

## Author Comment (AC1)

**Round 1**

**Overall Comments**

This manuscript addresses a scientifically important question about the sensitivity of Antarctic ice sheet projections to different sub-ice shelf melt rate treatments during model initialization. The core finding that the method of initialization of sub-ice shelf melt rates can lead to a significant difference in projected sea-level contributions is valuable for the ice sheet modeling community, and relevant to the scope of the journal.

**General Comments**

The research article presents novel insights by isolating the effects of sub-ice shelf melt rate treatment while maintaining identical model configurations, which addresses limitations of previous intercomparison studies that combined models of varying numerical complexities and initialization methods. The experimental design is well-motivated, and the writing is generally clear and well-structured.

The conclusions that West Antarctica dominates in driving projection uncertainties are potentially significant and well supported by the results, as are the substantial differences reported from previous studies. The comparison with CMIP6 and ISMIP6 projections raises important questions about model setup differences and other contributors beyond the melt rate parameterization that warrant further investigation. However, several methodological components require more detailed description to facilitate reproducibility, particularly the forward projection methodology, and the discussion of divergences from previous studies needs expansion.

**Response:** We thank the reviewer for the thoughtful and constructive comments, which have helped us significantly improve the manuscript. We are encouraged by the positive feedback on the novelty, experimental design, and significance of our findings. Below, we provide a point-by-point response to the specific suggestions raised.

1. In the model initialization, we have enhanced the description of both the similarities and differences between our model configuration and that of LOW21. To enhance the structural clarity of the manuscript, we hereafter refer to the initialization experiment using observed basal melt rates via Eq. 1 as experiment S1 (our study), and the experiment reproducing the LOW21 with Eq. 2 as experiment S2.

2. Furthermore, based on recent literature, we have provided additional justification

for using observed basal melt rates from Rignot et al. (2013). At coarse resolutions, the PISM utilizes a sub-grid grounding-line scheme to interpolate physical quantities within grid cells containing the grounding line. This approach ensures a smoother transition in the treatment of grounded and floating ice, resulting in a more accurate and physically realistic representation.

3. In the projection, based on whether the sub-grid scheme is enabled during simulation, we categorize the results into the more intuitive "sub-grid scheme on (SGO) scenario" and "sub-grid scheme off (SGF) scenario", replacing the original "high scenario" and "low scenario". Forthermore, we have added detailed information regarding the sources and temporal resolution of climate forcing.

4. Our results have been systematically compared against the ISMIP6 ensemble results, analyzing the strengths and limitations of our study. Furthermore, we have omitted the extrapolated AIS contribution beyond 2100 to maintain analytical rigor in the revised manuscript, as this projection was not substantiated by our simulation results.

5. We have added analysis on the key contributors to uncertainty in projections. Building on existing literature, we have also discussed potential future work directions aimed at better constraining the dominant drivers of model uncertainty. Additionally, we have analyzed the impact of errors in the observation for model validation. Specifically, we note that due to these uncertainties, the observed AIS state may potentially reflect an unstable condition, which could consequently affect the outcomes of model validation.

We have carefully addressed all the comments provided and believe the revised manuscript is substantially improved as a result. Once again, we thank the reviewers for their invaluable time and insightful suggestions.

**Specific Comments:**

**Methodological Details:**

**Sub-grid interpolation:**

**p5, l107:** *"... we conduct projection experiments, initiated in 2015, by employing "high" and "low" scenarios controlled by the sub-grid melt interpolation"*
This sub-grid melt interpolation requires clearer explanation. What constitutes this interpolation scheme, and why does activating it constitute a "high" scenario versus omitting it for "low" scenarios? Clarify the distinction between these scenarios.
**Response:** Thanks for your suggestions.

1. Simulating retreat processes of marine-terminating glaciers in coarse-resolution grid models, the sub-grid scheme calculates one-sided derivatives of the surface slope around the grounding line and interpolates key physical variables based on spatial gradients across the interface between grounded and floating cells. Assign 0 to ice-free/floating cells, 1 to fully grounded cells, and 0–1 to partially grounded cells (includes grounding line). The formula for basal melt rate adjusted using this scheme is:

$$M_{b,adjusted} = \lambda M_{b,grounded} + (1 - \lambda)M_{b,shelf\_base}$$

$M_{b,grounded}$, $M_{b,shelf\_base}$ denote the basal melt calculated for grounded ice grid cells and floating ice grid cells, respectively. $\lambda$ indicates the value (0-1) of the mask corresponding to the grid cell. This scheme is also used to adjust the basal friction in the transition zone. For more accurate expression, we have revised the term "sub-grid melt interpolation scheme" to "sub-grid grounding-line scheme" throughout the manuscript.

2. The "high scenario" and "low scenario" refer specifically to simulation results obtained by enabling or disabling the sub-grid scheme in PISM, respectively, and do not represent different climate (RCP/SSP) scenarios. For clarity, we have replaced the original terms "high scenario" and "low scenario" with the more descriptive expression "sub-grid scheme on (SGO) scenario" and "sub-grid scheme off (SGF) scenario". The "SGO scenario" activates the sub-grid melt interpolation, thereby accounting for basal melting in grid cells containing the grounding line. This results in higher overall mass loss because it accelerates grounding line retreat within the coarse-resolution grid; without it, the retreat would not be simulated. By contrast, the "SGF scenario" omits the scheme and applies no basal melting to any grid cell that is not entirely floating. These neglects melting in partially floating cells, leading to lower total mass loss and thus representing a more conservative (lower melt) scenario.

3. As noted in your comment on "**Projection uncertainties (p14, Table 2)**", limited computational resources prevented large ensemble simulations of AIS evolution for statistically significant projections. We therefore alternatively enabled/disabled this scheme to estimate the upper/lower bounds of the AIS sea-level contribution (Table 2).

4. As suggested, we have supplemented this section accordingly and modified the relevant description in the revision as follows:
   *"The grounding line migration is optimized through a sub-grid scheme, which calculates one-sided derivatives of the surface slope around the grounding line and interpolates key physical variables—such as basal shear stress, basal melt rate, and basal friction—based on spatial gradients across the interface between grounded and floating cells (Feldmann et al., 2017; Nowicki et al.,2020). This approach reduces physical gradients across the grounding line and simulates a more realistic and dynamic representation of the ice margin (Leguy et al., 2014; Golledge et al., 2015).".*

*"Further, based on the initialized model state and the optimal parameter set from S1, we conduct projection experiments from 2015 by turning on or off the sub-grid grounding-line scheme in PISM. The "sub-grid scheme on (SGO) scenario" incorporated sub-grid melt interpolation near grounding lines, accelerating grounding-line retreat in our coarse-resolution model, while the "sub-grid scheme off (SGF) scenario" ignored melt in partially floating cells, yielding more conservative mass loss estimates (Albrecht et al., 2011; Golledge et al., 2015; Nowicki et al.,2020).".*

**Climate forcing implementation:**

**p5, l109-110:** *"These experiments used climate forcing derived from the CMIP5 IPSL-CM5A-MR (Barthel et al., 2020; Payne et al., 2021) and the CMIP6 CNRM-CM6-1 (Nowicki et al., 2016; Kamworapan et al., 2021) to assess ..."*

I could also not find details on the implemented climate forcings, consider specifying:

-How are the forcing fields (temperature, salinity?) used/derived from these models?

-What is the temporal resolution of the forcing data?

-How are the CMIP5 or CMIP6 forcings interpolated or downscaled to the PISM grid?

Describing this would be useful for reproducibility

**Response:** Thanks for your suggestions.

1. We directly utilized the "ISMIP6 21st Century Forcing Datasets" published by Nowicki et al. (2021), which provide 21st-century atmospheric and oceanic forcing datasets designed for standalone ice sheet model simulations over Greenland and Antarctica. The dataset incorporates output from six CMIP5 and four CMIP6 climate models, processed into a form readily applicable to ice sheet models. Each climate model contributes atmospheric forcing—including surface mass balance anomaly and near-surface air temperature anomaly—from 1995 to 2100, as well as oceanic forcing covering the same period, which includes salinity, temperature, and thermal forcing. So we can directly download the output forcing fields from these different climate models without the need for any additional processing.

2. The original temporal resolution of the atmospheric and oceanic forcing data is "daily". However, as PISM is not suited for high-temporal-resolution inputs, we pre-process the data into annual averages before using them in the model. Spatial grid resolutions are available in 2 km, 4 km, 8 km, 16 km, and 32 km, can be selected as needed.

3. PISM provides publicly available preprocessing scripts (https://github.com/pism/pism-ais) to convert model input data into a consistent resolution and a model-readable format. For our experiments, the input data were standardized to an 8 km grid resolution. Therefore, the 8 km horizontal resolution

forcing data can be downloaded directly from the "ISMIP6 21st Century Forcing Datasets" without the need for re-interpolation.

4.  As suggested, we have added the relevant content in the manuscript and supplemented the source description of the forcing data in the Data Availability Statement:

*"We employed the same daily-resolution climate forcing as LOW21 (Lowry et al., 2021), derived from the CMIP5 IPSL-CM5A-MR RCP2.6/8.5 (Barthel et al., 2020; Payne et al., 2021; Nowicki et al., 2021) and the CMIP6 CNRM-CM6-1 SSP1-2.6/5-8.5 product (Nowicki et al., 2016; Kamworapan et al., 2021; Nowicki et al., 2021) spanning 2015–2100, to assess and compare Antarctica's contribution to global mean sea-level rise by 2100.".*

*"The forcing data under RCP and SSP scenarios were sourced from the dataset published by Nowicki et al. (2021). The data preprocessing tool used is the publicly available scripts pism-ais (https://github.com/pism/pism-ais).".*

**Model configuration:**

**p11, l226:** *"consistent model configurations and climate forcings with LOW21.."*

It is not clear what consistent model configurations constitutes of, and the term is ambiguous. There is some clarification for this provided in Section 5, which would fit better in Section 2, where the experimental setup is first introduced. Consider establishing early on exactly which parameters remain identical between experiments and which differ.

**Response:** Thanks for your suggestions.

1. During initialization, our model configuration—including parameters, stress approximation, resolution, initial topography, and atmospheric conditions—is identical to LOW21. The only difference lies in the oceanic initial condition: our simulation uses observationally derived sub-ice-shelf melt rates, whereas LOW21 employed ocean temperature and salinity. To enhance the structural clarity of the manuscript, we hereafter refer to the initialization experiment using observed basal melt rates via Eq. 1 as S1, and the experiment reproducing the LOW21 with Eq. 2 as S2. In the projection experiments, both the model configuration and climatic forcing are the same.

2. Per your suggestions, we have clarified differences in the initialization and projection experiments and moved model configuration details from Section 5 to Section 2. We have revised the relevant description as follows:

*"During initialization procedure, to evaluate the specific role of oceanic conditions, we conducted two experiments using PISM: Experiment "S2" replicates the single simulation from LOW21 that used the best-fit parameter set (the one minimizing*

*mismatch with observations), employing a thermodynamic parameterization (Eq. 2) to estimate sub-ice shelf melt rates. Experiment "S1" uses the same model configuration—including all parameters, stress balance approximation, resolution, topography, and atmospheric conditions—but replaces the basal melting scheme with observed basal melt rates derived from satellite altimetry (ICESat-1), radar (OIB and ALOS PALSAR), and model outputs (RACMO2), based on Eq. 1.".*

*"We employed the same daily-resolution climate forcing as LOW21 (Lowry et al., 2021), derived from the CMIP5 IPSL-CM5A-MR RCP2.6/8.5 (Barthel et al., 2020; Payne et al., 2021; Nowicki et al., 2021) and the CMIP6 CNRM-CM6-1 SSP1-2.6/5-8.5 product (Nowicki et al., 2016; Kamworapan et al., 2021; Nowicki et al., 2021) spanning 2015–2100, to assess and compare Antarctica's contribution to global mean sea-level rise by 2100.".*

**Forward projection:**

**p12, l231:** *"Prognostic simulations from 2015 to 2100 revealed divergent ice mass changes compared to LOW21, particularly in WAIS. Under various climate scenarios, ..."*

The treatment of sub-ice shelf melt rates during the 2015-2100 projection period is unclear. Are the observational melt rates from Rignot et al. (2013) prescribed throughout the projections?

**Response:** Thanks for your suggestions.
1. The sub-ice shelf melt rates from Rignot et al. (2013) were used solely during the initialization in S1 to construct a new ice-sheet initial state under this oceanic condition, enabling comparison with S2 (LOW21) and analysis of resultant dynamic mechanism differences.
2. Furthermore, our study focuses on how variations in sub-ice shelf melt rates affect the initial ice-sheet state after spin-up and subsequently influence projected sea-level contributions. To maintain consistency with the LOW21 for comparative analysis in projection experiment, we used the same future oceanic forcing—specifically ocean temperature and salinity from CMIP5 and CMIP6 climate models—to project future ice-mass change.
3. To improve clarity, we have revised the expression to *"To ensure that differences in projections originated solely from the model spin-up, the basal melting scheme was parameterized using the same linear thermodynamic framework for the ice-shelf–ocean boundary layer as that employed in LOW21. This approach explicitly resolves heat and freshwater exchange processes at the ice–ocean interface, driven by oceanic forcing under different RCP/SSP scenarios from 2015 to 2100.".*

**p14, Table 2:**

220 The confidence intervals presented in Table 2 are not defined. Are these ranges derived from ensemble runs, sensitivity tests, or are a statistical treatment of the model output? Consider clarifying the source and methodology of these intervals and expand the table description/title accordingly

**Response:** Thanks for your suggestions.

225 1. The confidence intervals provided in Table 2 summarize simulation results obtained by either enabling or disabling the sub-grid grounding-line scheme in the prediction process. When using the sub-grid scheme, the model applies weighted adjustments to ice mass changes in the ice sheet–shelf transition zone. This results in the "SGO scenario", which defines the upper bound of the confidence interval for the sea-

230 level contribution in Table 2. Conversely, when this scheme is disabled, the model neglects these ice mass changes, yielding the "SGF scenario" that defines the lower bound of the interval. This methodology follows the approach of Golledge et al. (2015).

2. In response to your feedback, we have supplemented the description of Table 2:

235 *"Sea level contribution (m SLE) of Antarctic Ice Sheet Basins by 2100. The confidence intervals represent the range of sea-level contribution from the "SGO scenario" to the "SGF scenario" simulation across different RCP/SSP scenarios; the single value denotes the mean value of this range.".*

**Initialization results:**

240 **Grounding line migration:**

**p11, l225:** *"...discrepancy can be attributed to the reversibility of grounding line migration on a retrograde-slope bedrock, which is characterized by oscillatory shifts"*

The differences presented among the three regions of retrograde bed slope (TB, WL, GVL) are interesting, and Section 3.3 provides a well written mechanistic

245 understanding of the processes involved. In Section 3.4, the grounding line analysis mentions "reversibility of grounding line migration on retrograde-slope bedrock", raising an important point, but remains brief. Further, the phrasing of that sentence makes it unclear, as it seems to conflate the mechanism of oscillatory shifts in the grounding line with the bed geometry. A clearer explanation of the physical processes underlying these patterns in GL migration would be useful.

250 underlying these patterns in GL migration would be useful.

**Response:** Thanks for your suggestions.

1. In Section 3.4, the physical mechanism behind grounding line retreat involves higher melt rates triggering MISI, leading to sustained retreat on a retrograde slope. This also highlights the role of bed topography in grounding line migration. So, the subsequent paragraph further emphasizes that the factors of grounding line motion, such as topography, ice velocity, and basal melt rates. Grounding line migration can be driven by either individual or combined factors. For example, the discrepancy in grounding line position on the Siple Coast is primarily influenced by topography (Figs. 3c, d), as evidenced by the agreement between S1 and S2 (Figs. 5d, e). In contrast, in the TB of WAIS (Section 3.4), grounding line retreat results from the combined effects of enhanced basal melt and retrograde bed topography.

2. The original manuscript did not clearly describe the cause of grounding line migration on the Siple Coast; the explanation was unclear and potentially confusing. Therefore, incorporating your feedback, we have revised the explanation to clarify why the simulated grounding line position on the Siple Coast is located closer to the open ocean compared to observational data:

   *"In fact, the grounding line position varies across different ice streams depending on topography, neighboring ice shelf basal melt rates, and ice velocities (Martin et al., 2011).".*

   *"This discrepancy likely arises from the stabilizing self-limiting mechanism inherent to prograde slopes. As the grounding line retreats into shallower bedrock, the ice thins and flux decreases; this leads to ice re-accumulation that prompts grounding line readvancement, creating reversible shifts around an equilibrium point (Huybers et al., 2017).".*

**Projection results:**

**p12, l238-239:** *"which is relative to the hysteretic response of ice sheet dynamics to climate forcing…"*

The lack of scenario dependence with significant overlap in the prediction ranges is consistent with the delayed/hysteretic response of ice sheet dynamics in that the current ice sheet state and near future (as during this 2015-2075 period) reflect historical forcing, but it is unclear from this sentence. Consider explaining/rephrasing

**Response:** Thanks for your suggestions. This sentence indeed did not clearly explain the reason for the overlapping projections across scenarios before 2075. This is because the ice sheet's hysteretic response means that its full reaction to new climatic forcings takes considerable time to appear. Therefore, changes during this period primarily

reflect the ice sheet's response to past historical forcing, rather than to divergent future emission scenarios. In response to your comment, we have revised the statement as follows:

*"This is consistent with the hysteretic response of ice sheet dynamics, meaning that the ice sheet's state in the near-term (2015-2075) is largely determined by historical forcing, masking the influence of divergent future scenarios (Garbe et al., 2020).".*

**p12, l244-246:** *"...2100 trajectory extensions, persistent ice mass ....beyond 2100. "*

Are these extended projections beyond 2100 provided anywhere for this particular study, or does this statement refer existing literature? Consider expanding on this, particularly, which SSP scenario and which RCP simulations correspond to persistent ice mass and stabilizing trends.

**Response:** Thanks for your suggestions. We did not conduct extended projection experiments of ice sheet evolution beyond 2100, as climate forcing datasets after 2100 are not publicly available in the "ISMIP6 21st Century Forcing Datasets". As such, we have removed the corresponding speculative statement from the manuscript to maintain rigor. All analyses and conclusions in the revised text are now strictly based on simulations ending in 2100.

**p15, l295-300:** *"... Compared to ISMIP6 (Ice Sheet Model Intercomparison for CMIP6) Antarctic projections under RCP 8.5 (Seroussi et al., 2020), our WAIS contribution exceeds it by approximately 0.15 m SLE, with AP showing a slight increase (~0.002 m SLE) and EAIS exhibiting a minor reduction (~0.02 m SLE...."*

The substantial differences from ISMIP6 projections raises important questions. To strengthen this section, please clarify whether the comparisons are made against the full ensemble or PISM-based runs alone. (e.g., from the description in LOW21, the model setup here would be most comparable to the VUW-PISM from ISMIP6). It would also be helpful to highlight and discuss any other model differences apart from the melt rate parameterization that could contribute to the divergences. If possible, a short comparison on the strengths/limitation of either this observationally constrained approach or the ISMIP6 approach would add significant value.

**Response:** Thanks for your suggestions.

1. The comparisons presented in this section are made against the full ensemble simulated results of the ISMIP6 projections (Seroussi et al., 2020), rather than against any single model. We did not perform a direct comparison solely against the PISM-based simulated results from ISMIP6 because the sea-level contribution projections for individual ice sheet models were not separately provided in the main

ISMIP6 ensemble publications. The published results primarily offer the multi-model ensemble results, which formed the basis for our comparative analysis.

2. The uncertainties in the projections are related to the physical processes represented in the model, the model initial conditions, the forcing data, and the model parameterization schemes. We have addressed these aspects in Section 5 ("Model Uncertainties") and have revised the discussion in response to your comments.

3. The ISMIP6 ensemble, which combines ice-flow model simulations from 13 international groups, provides a more comprehensive representation of the full spectrum of potential AIS behaviors under given climatic forcing. Results indicate that, among the three major sources of uncertainty in sea-level contribution, the parameterization of oceanic conditions into basal melt rates is the dominant contributor. However, the ensemble-based simulated results inherently cannot reveal the specific physical mechanisms driving these differences. A limitation of our single-model study is its dependency on the parameterizations of the PISM, which can only represent a subset of potential future sea-level contributions and do not provide statistically robust uncertainty ranges. However, by following the same single-model approach and climatic forcing as LOW21 ensemble experiments, and by comparing simulations using both observationally derived basal melt rates and parameterized ocean thermal forcing, S1-based projection identifies the specific regions and dynamic mechanisms underlying the ISMIP6 finding that "the parameterization of ocean thermal forcing into basal melt rates is the largest source of projection uncertainties".

4. As suggested, we have included a short section comparing the strengths and limitations of our approach versus the ISMIP6 methodology. The content we have supplemented is as follows:

*"Compared to the full ensemble results of ISMIP6 (Ice Sheet Model Intercomparison for CMIP6) Antarctic projections under RCP 8.5 (Seroussi et al., 2020), S1 simulated sea-level contribution from WAIS is approximately 0.15 m SLE higher, while AP showing a slight increase (~0.002 m SLE) and EAIS exhibiting a minor reduction (~0.02 m SLE).".*

*"The ISMIP6-Antarctica projections improve a more comprehensive representation of potential Antarctic sea-level contribution under climatic forcing, with the parameterization of oceanic conditions into basal melt rates being the dominant source of uncertainty (Seroussi et al., 2020). However, this approach cannot identify the specific physical mechanisms behind the inter-model differences. A key limitation of our single-model research is its reliance on PISM-specific parameterizations, which restrict the range of projected sea-level contributions and*

*provide limited statistical uncertainty. By comparing observationally derived and parameterized basal melt rates under a consistent single-model framework, our simulations identify the specific regions and dynamic mechanisms underlying the ISMIP6 projection uncertainty associated with representation of oceanic conditions.".*

**p16, l318-321:** *"Compared to other prior studies, our sea level projections differ due to variations in ice sheet model configurations, including model resolution, ice dynamics (particularly stress balance schemes), represented physical processes (calving, hydrology, or bedrock uplift), and initialization methods (data assimilation or spin-up)"*

These lines address my above comment about detailing the potential reasons for the differences. Is there evidence to suggest which of these factors might be most important, and how it could be addressed in future work?

**Response:** Thanks for your suggestions.

1. According to the ISMIP6-Antarctica projections (Seroussi et al., 2020), the parameterization of ice melt dynamics contributes most significantly to the uncertainty in sea-level estimates, surpassing variations arising from differences in climate model forcing, initialization methods, and the physical processes included. This implies that ice-model-related uncertainties dominate throughout the simulation period (Seroussi et al., 2019; Seroussi et al., 2023).

2. Therefore, continual model improvement, further exploration of the broader parameter space covered by initial state ensembles and their extended sampling (Coulon et al., 2024; Klose et al., 2024), and the acquisition of more observations for verification and validation are essential to reduce uncertainties in future projections of dynamic mass loss from the AIS (Favier et al., 2019; Seroussi et al., 2020; Seroussi et al., 2023).

3. As suggested, we have incorporated this content as follows:

   *"Of these factors, the parameterization of ice melt dynamics contributes most significantly to the uncertainty in sea-level estimates, surpassing uncertainties arising from differences in climate model forcing, initialization methods, and the selected physical processes. This implies that ice-model-related uncertainties dominate throughout the simulation period (Seroussi et al., 2019, 2023). Therefore, continual model improvement, further exploration of the broader parameter space covered by initial state ensembles, and its extended sampling are essential to reduce uncertainties in future projections of dynamic mass loss from the AIS (Favier et al., 2019; Coulon et al., 2024; Klose et al., 2024).".*

**Technical Comments:**

**p1, l30-31:** This sentence on ice shelf susceptibility could be streamlined for clarity.

**Response:** Thanks for your suggestion. We have revised it as follows: *"Ice shelves are highly vulnerable to oceanic forcing due to both basal melting from warm seawater and their near-flotation elevations (Bindschadler et al., 2013; Depoorter et al., 2013; Li et al., 2023)."*.

**p3, l74:** "(Rignot et al., 2013; Fig. 2; Table 1)" Should this reference be "Fig. 1"?

**Response:** Thank you for your suggestion and correction. Yes, we have changed "Fig. 2" to "Fig. 1".

**p3, l90:** What is the depth of the ocean water temperature, T_s? Is it taken close to the bottom?

**Response:** Thanks for your suggestion. We have supplemented the definition of T_s according to the relevant literature: *"$T_s$ is the vertically averaged ocean temperature between 200 m and 1000 m depth along the continental slope (assigned $T_s$=271.45 K, Beckmann and Goosse, 2003; Martin et al., 2011)"*.

**p3, l92:** Define S_o (ocean salinity?) in Equation (3)

**Response:** Thanks for your suggestion. We have supplemented the definition of S_o: *"$S_0$ denotes the specified ocean salinity (35 psu)."*.

**p8, l166-170:** In Fig. 5 caption, clearly define ΔRMSE (Is it RMSE (current vs. obs.) – RMSE (LOW21 vs. obs.)?)

**Response:** Thanks for your suggestion. Yes, ΔRMSE is the result of RMSE (current vs. obs.) – RMSE (LOW21 vs. obs.). We have added a calculation formula to clarify the method for determining ΔRMSE: *"$\Delta RMSE = RMSE_{S1\ (our\ study)} - RMSE_{S2\ (LOW21)}$"*.

**p9, l175:** The percentages mentioned (0.5%, 0.2%) appear to conflict with the values presented in Table 1. Clarify whether these refer to overall ice volume biases or volume above flotation biases.

**Response:** Thanks for your suggestion.

1. The value of 0.5% represents the difference between the bias in S1 simulated ice volume above flotation relative to observations (-4.61%) and the bias in S2 simulation relative to observations (-5.14%), calculated as -4.61%-(-5.14%) =0.53%. And the 0.2% corresponds to the difference in the simulation results for

the GVL region: -5.23%-(-5.42%) =0.19%. (Note: The final values 0.53% and 0.19% are retained for accuracy, while the text acknowledges the rounded reference "0.5%" and "0.2%" from the original context.)

2. The term "volume" refers to volume above flotation biases. We have revised the statement to: *"the bias in ice volume above flotation decreases by approximately 2.8%, while the biases for WL and GVL reduced by 0.5% and 0.2% (Table 1), respectively."*.

**p9, l186:** "enhanced oceanic forcing" needs a clearer definition in this context.

**Response:** Thanks for your suggestion. We have modified the relevant expressions in the revision as follows: *"In this study, enhanced oceanic forcing (Fig. 2), which is represented by higher basal melt rates, intensifies ice-shelf basal melting, leading to geometric thinning and reduced buttressing effect of upstream ice flow (Gudmundsson, 2013; Miles et al., 2022)."*.

**p9, l195:** "elevated basal water consent": Should be "…content"

**Response:** Thank you for your suggestion and correction. We have changed "consent" to "content".

**p11, l211-212:** Line colors mentioned here are inconsistent with Fig. 7 description. (purple/orange in text vs. grey (dashed/solid) in figure)

**Response:** Thank you for your suggestion. We have revised the description in the text based on Fig. 7: *"Cross-sectional analysis of Thwaites Glacier (Fig. 7) demonstrates this mechanism, with enhanced basal melting, causing an approximately 30 km grounding line retreat from its stabilized position (S2, dashed grey line in Fig. 7) to a new quasi-stable state (S1, solid grey line in Fig. 7)."*.

**p12, l243:** "persistent ice mass": Should this be "ice mass loss"?

**Response:** Thank you for your suggestion. Based on your comment in the **Projection results (p12, l244-246)**, we have revised the entire paragraph by replacing "ice mass loss" with "*the AIS contribution*".

**p12, l243-247:** There seems to be a causal gap between the two sentences. Consider adding a brief description in between to establish the causal link between persistent ice mass (loss?) / stabilizing trends and "amplified ice-climate feedback" or modifying the sentence.

**Response:** Thank you for your suggestion. We have modified the description and logic of this passage as follows: *"These differences between RCP 8.5 and SSP 5-8.5*

*projections are largely due to the SSP scenarios in CMIP6 climate models simulate higher warming magnitudes (averaging +0.14-0.25 ℃) than RCP scenarios in CMIP5 at equivalent radiative forcing (Tokarska et al., 2020; Wyser et al., 2020; Rounce et al., 2023). Consequently, under anthropogenic warming, the sea-level commitment of AIS under SSP high-risk scenarios demands heightened scientific attention.".*

**p12, l250:** Fig. 8 caption should specify which RCP scenarios are used for panels (a) and (b).

**Response:** Thank you for your suggestion. Fig. 8 (a) shows the spatial differences in projected mean ice thickness based on S1 and S2 under RCP 2.6 and 8.5 scenarios, while (b) displays the spatial differences of the ensemble mean for the year 2100. We have revised the description in Fig. 8: *"Spatial differences in the projected mean ice thickness between the multi-scenario (RCP 2.6 and RCP 8.5) ensemble means from S1 and S2 in 2050 (a) and 2100 (b).".*

**Secondary comments:**

The below comments are merely secondary suggestions for the authors, and I leave it to them whether to address these or not.

**Use of Rignot et al., (2013) melt rates:**

The reliance on Rignot et al., (2013) melt rate observations is understandable given their wide use in the modeling community. At the same time, this data now reflects conditions around a decade old, and more recent work, has further revealed interannual variability in ice shelf basal melt rates (Adusumilli et. al., 2020), and even slowdown in melt-driven thinning for certain sectors (Paolo et. al., 2023). I encourage the authors to briefly discuss and frame the use of this dataset within the context of more recent observational work.

**Response:** Thanks for your suggestion. We agree that acknowledging the temporal limitations of the Rignot et al. (2013) melt rate dataset is important.

1. We clarify that the melt rates from Rignot et al. (2013) were adopted herein because they align more closely with the ocean thermal forcings in the Schmidtko et al. (2014) dataset (1975–2012), as used in experiment S2. We note, however, that these rates represent a six-year mean (2003–2008) from ICESat observations and thus do not capture interannual variability or more recent changes, as highlighted in Adusumilli et al. (2020) and Paolo et al. (2023). So we consider experiment S1 use of this dataset as a general representation of long-term mean ice-shelf basal melt

490 conditions, while also emphasizing the need for future work to incorporate time-varying melt forcings to better understand ice-ocean interactions.

2. In response, we have added a discussion paragraph in the manuscript:

*"It is important to note that the ice-shelf basal melt rates applied here, derived from Rignot et al. (2013), were selected for use because the ocean thermal forcing they*
495 *represent corresponds closely to the 1975–2012 mean state of the Southern Ocean captured in Schmidtko et al. (2014) dataset (used in S2). However, as these data reflect conditions from approximately a decade ago, they inherently represent a temporal average and do not capture interannual variability in ocean forcing (Adusumilli et al., 2020). Furthermore, Paolo et al. (2023) observed a widespread*
500 *slowdown in ice-shelf thinning across the Amundsen, Bellingshausen, and Wilkes sectors, attributing it to changes in ocean forcing and internal ice-dynamic feedbacks. Therefore, S1 simulated results should be interpreted as a response to a steady-state, general ice-shelf basal melting field. Future work would benefit from incorporating time-evolving melt rates to better constrain the sensitivity of the AIS*
505 *to oceanic variability on interannual to decadal timescales.".*

**Validation against observations:**

**p16, l321:** *"...the present-day AIS may not have been in a steady-state during the observational period, and thus, some of the misfits could be attributed to uncertainties in the observational data used for validation"*

510 The authors note that the present-day AIS may not have been in steady state during the observational period. A brief discussion of the uncertainties and potential biases in the datasets used for validation would strengthen this argument

**Response:** Thanks for your suggestions.

1. We have supplemented the discussion on the uncertainties and potential biases in the observational data used for validation. As noted in the context of BedMachine
515 v.3 (Morlighem et al., 2019), approximate calculations or inversion methods introduce errors across various regions—including fast-flowing sectors, slow-moving zones, ice-free land, ocean bathymetry, and sub-ice-shelf cavities—with estimated biases ranging between 10 and 30 m depending on the area. Similarly,
520 for the MEaSUREs velocity product (Mouginot et al., 2019), ice surface velocity inevitably incorporates errors arising from speckle-tracking and phase data during SAR data processing, along the direction of ice flow. These inherent errors and biases in the observed ice sheet state—which itself may not be in steady state—contribute to the apparent mismatch between model simulations and observational
525 data.

2. As suggested, we have expanded the discussion on ice sheet non-steady-state behavior in the revision. *"Notably, the present-day AIS may not have been in a*

*steady-state during the observational period (Martin et al., 2011). This inference, while primarily based on discrepancies between model simulations and observations, may also be influenced by uncertainties inherent in the validation datasets. For example, the BedMachine v3 dataset relies on approximate calculations in regions such as ice-free land, ocean bathymetry, and cavities under ice shelves, potentially introducing spatial biases in thickness estimates (Morlighem et al., 2019). Similarly, the MEaSUREs velocity map inevitably contains errors in flow direction derived from phase data and speckle tracking during SAR data processing (Mouginot et al., 2019). Thus, the apparent model– data mismatch not only demonstrates the non-steady-state of AIS but also reflects the challenge of validating model simulations against modern records that contain their own uncertainties and potential biases.".*

Paolo et al., 2023, which constructed 3 km resolution datasets of ice thickness also revealed a slowdown in thinning from around 2008, specifically in the Amundsen, Bellingshausen and Wilkes sectors. Following from my earlier comment about melt rate observations, in addition to the citations in the introduction for accelerated thinning of ice shelves, this article may be worth including as a reference. The specific paper I'm referring to is:

Paolo, F. S., Gardner, A. S., Greene, C. A., Nilsson, J., Schodlok, M. P., Schlegel, N.-J., and Fricker, H. A.: Widespread slowdown in thinning rates of West Antarctic ice shelves, The Cryosphere, 17, 3409–3433, https://doi.org/10.5194/tc-17-3409-2023, 2023.

**Response:** Thank you for your suggestion. We agree that the study by Paolo et al. (2023) provides a highly relevant and updated observational context for the discussion of ice shelf basal melt rates and their variability. We have now included a citation to this important work in the revised manuscript. This addition strengthens our discussion on the recent temporal variability in ice-shelf basal melting.

**References:**

Adusumilli, S., Fricker, H. A., Medley, B., Padman, L., & Siegfried, M. R. (2020). Interannual variations in meltwater input to the Southern Ocean from Antarctic ice shelves. *Nature Geoscience*, *13*(9), 616-620. https://doi.org/10.1038/s41561-020-0616-z

Favier, L., Jourdain, N. C., Jenkins, A., Merino, N., Durand, G., Gagliardini, O., Gillet-Chaulet, F., & Mathiot, P. (2019). Assessment of sub-shelf melting parameterisations using the ocean–ice-sheet coupled model NEMO(v3.6)– Elmer/Ice(v8.3). *Geoscientific Model Development*, *12*(6), 2255-2283. https://doi.org/10.5194/gmd-12-2255-2019

Feldmann, J., Albrecht, T., Khroulev, C., Pattyn, F., & Levermann, A. (2017).

Resolution-dependent performance of grounding line motion in a shallow model compared with a full-Stokes model according to the MISMIP3d intercomparison. *Journal of Glaciology*, *60*(220), 353-360. https://doi.org/10.3189/2014JoG13J093

Huybers, K., Roe, G., & Conway, H. (2017). Basal topographic controls on the stability of the West Antarctic ice sheet: lessons from Foundation Ice Stream. *Annals of Glaciology*, *58*(75pt2), 193-198. https://doi.org/10.1017/aog.2017.9

Leguy, G. R., Asay-Davis, X. S., & Lipscomb, W. H. (2014). Parameterization of basal friction near grounding lines in a one-dimensional ice sheet model. *The Cryosphere*, *8*(4), 1239-1259. https://doi.org/10.5194/tc-8-1239-2014

Nowicki, S., Goelzer, H., Seroussi, H., Payne, A. J., Lipscomb, W. H., Abe-Ouchi, A., Agosta, C., Alexander, P., Asay-Davis, X. S., Barthel, A., Bracegirdle, T. J., Cullather, R., Felikson, D., Fettweis, X., Gregory, J. M., Hattermann, T., Jourdain, N. C., Kuipers Munneke, P., Larour, E., Little, C. M., Morlighem, M., Nias, I., Shepherd, A., Simon, E., Slater, D., Smith, R. S., Straneo, F., Trusel, L. D., van den Broeke, M. R., & van de Wal, R. (2020). Experimental protocol for sea level projections from ISMIP6 stand-alone ice sheet models. *The Cryosphere*, *14*(7), 2331-2368. https://doi.org/10.5194/tc-14-2331-2020

Nowicki, S., Simon, E., & ISMIP6 Team. (2021). ISMIP6 21st Century Forcing Datasets [Data set]. The Ghub. https://doi.org/10.5281/zenodo.11176009

Paolo, F. S., Gardner, A. S., Greene, C. A., Nilsson, J., Schodlok, M. P., Schlegel, N.-J., & Fricker, H. A. (2023). Widespread slowdown in thinning rates of West Antarctic ice shelves. *The Cryosphere*, *17*(8), 3409-3433. https://doi.org/10.5194/tc-17-3409-2023

Seroussi, H., Verjans, V., Nowicki, S., Payne, A. J., Goelzer, H., Lipscomb, W. H., Abe-Ouchi, A., Agosta, C., Albrecht, T., Asay-Davis, X., Barthel, A., Calov, R., Cullather, R., Dumas, C., Galton-Fenzi, B. K., Gladstone, R., Golledge, N. R., Gregory, J. M., Greve, R., Hattermann, T., Hoffman, M. J., Humbert, A., Huybrechts, P., Jourdain, N. C., Kleiner, T., Larour, E., Leguy, G. R., Lowry, D. P., Little, C. M., Morlighem, M., Pattyn, F., Pelle, T., Price, S. F., Quiquet, A., Reese, R., Schlegel, N.-J., Shepherd, A., Simon, E., Smith, R. S., Straneo, F., Sun, S., Trusel, L. D., Van Breedam, J., Van Katwyk, P., van de Wal, R. S. W., Winkelmann, R., Zhao, C., Zhang, T., & Zwinger, T. (2023). Insights into the vulnerability of Antarctic glaciers from the ISMIP6 ice sheet model ensemble and associated uncertainty. *The Cryosphere*, *17*(12), 5197-5217. https://doi.org/10.5194/tc-17-5197-2023

---

## Author Comment (AC2)

**Round 1**

**Overall Comments**

This manuscript assesses how different sub-shelf melting schemes used to spin up ice-sheet models lead to similar but not identical initial ice-sheet states. This impacts considerably projections of the Antarctic ice sheet's contribution to future sea-level rise. The issue of initialization was treated in part by initMIP-Antarctica but given that different models with different setups were used within that project, the specific effect of different initial conditions was not isolated. This study is interesting because it represents a further step in this direction. A better characterization of initial conditions should help to reduce uncertainties in projections. I think this result deserves to be conveyed. However, I have a few major issues that prevent me from accepting this paper at this stage.

First, the methodological approach needs to be explained in more detail. Right now it is unclear to me what experiments have been done: have the authors used the data from Lowry et al (2021) for their comparison or have they redone simulations using the same basal melting scheme as in that study? This is critical because to be sure that the results shown are only due to the different sub-shelf melting initialization the experiments would need to be redone with exactly the same model version and configuration.

**Response:** We thank the reviewer for thoughtful and constructive comments, which have helped us significantly improve the manuscript.

1. Based on the publicly available input data and scripts from Lowry et al. (2021), we reproduced one set of their experiments using the same ice-sheet model (PISM v.1). Lowry et al. (2021) selected four key parameters controlling ice-sheet dynamics in PISM (sia_e, ssa_e, q, phi) for parameter optimization and identified the combination yielding the smallest deviation from observation (2.4, 0.6, 0.25, 10). To ensure the successful replication of the simulation results using this optimal parameter set, we engaged in extensive email correspondence with Prof. Lowry. Through multiple rounds of communication, he provided detailed guidance on parameter configuration and result validation, enabling us to accurately reproduce the experiments.

2. Upon reproducing the complete and systematic optimal parameters configuration from Lowry et al. (2021), we conducted another simulation using the same input data and atmospheric conditions, but applying different oceanic boundary conditions derived from observed sub-ice shelf basal melt rates. This approach

35  ensures that any differences in initial state originate solely from variations in oceanic conditions, rather than other factors.

3.  Following your suggestion to enhance the manuscript's structural clarity, we now refer to the initialization experiment using observed basal melt rates via Eq. 1 as "S1", and the experiment reproducing LOW21 using TF-linear parameterization

40  via Eq. 2 as "S2". This new term has been used consistently throughout the revised manuscript. As shown in Fig. R1a, S2 illustrates the process of reproducing these optimal parameter-based simulations of LOW21.

4.  The study by Lowry et al. (2021) did not utilize a scheme based on observed sub-ice shelf basal melt rates.

[Figure]

**Figure R1: Overview of model initialization and projection.** The schematic summarizes the model setup during spin-up (grey box) and projection (blue box), using observed basal melt rates together with ice-shelf basal temperature (S1, orange box) and Southern Ocean temperature and salinity (S2, yellow box; LOW21).

50   Also, the basal melting scheme used in the projections is not clearly described. The authors should clarify this and also show how the basal melting fields evolve smoothly from the spinup into the projections.

**Response:** Thanks for your suggestions.

1. Our study aims to investigate the impact of sub-ice shelf melt rates during model
55   initialization and to examine how the new initial state influences future ice-sheet evolution. To investigate the impacts of model initialization on projections, we maintained the same climate forcing (RCP/SSP) and basal melting scheme (TF-linear parameterization) as one set of projections in LOW21. In other words, observed basal melt rates were not used during the projection (Fig. R1b).

60   2. The new initial state—generated from those observed melt rates via spin-up—provided the initial ice-sheet geometry as the initial values for the projections. Atmospheric and oceanic boundary conditions were prescribed by RCP/SSP scenarios time series, replacing the constant oceanic conditions used in spin-up.

3. During the projection, all results—including basal melting fields—are recalculated
65   by solving the ice dynamics code using these initial values under the new climate forcing scenarios. This ensures a physically consistent and smooth transition between the spin-up and projection simulations.

Finally, the section on "Model Projection Results" lacks proper illustration of some of the results that are discussed, including the comparison to LOW21 and the descriptions
70   of relevant physical mechanisms that are described but are not illustrated, so that they remain speculative.

**Response:** Thanks for your suggestions.

1. In the revised manuscript, we have added a comparison of the AIS sea-level commitment based on projections from the S1 and S2 initial states (Figs. 8, 9). The
75   S1-based projection simulates future Antarctic evolution under RCP/SSP scenarios, starting from a present-day ice-sheet state obtained through spin-up using observed basal melt rates (S1). The S2-based projection utilizes the set of sea-level contribution results provided by Prof. Lowry, which were generated using the same parameter configuration as our S2 reproduction initialization. To ensure
80   comparability, we used the same climate forcing scenarios as in LOW21 for both projections.

2. As you mentioned, our simulations do not provide direct evidence (such as figures or tables) to conclusively demonstrate that certain physical mechanisms are the dominant processes behind the results. Therefore, we have adjusted the logical

85     framing of these discussions, transitioning from statements of "fact" to "well-supported inferences" based on existing literature, thereby avoiding any expression of speculation.

Another less critical issue is that several figures do not appear in the order in which they are cited.

90 **Response:** Thanks for your suggestions. We have adjusted the positioning of the figures in response to your specific comments, which has strengthened the logical flow of the manuscript.

We have carefully addressed all the comments provided and believe the revised manuscript is substantially improved as a result. Once again, we thank the reviewer for 95 the invaluable time and insightful suggestions.

**Specific Comments (numbers indicate the lines to which the comments refer):**

Title: "Antarctica ice sheet" should be "the Antarctic ice sheet" here and everywhere below.

**Response:** Thanks for your suggestions. We have made the corresponding changes — 100 8 in total across the manuscript.

**Abstract**

**15:** "impedes" sounds too strong; I suggest "brakes" or "reduces". Also, what does "the state of sub ice-shelf melting" refer to?

**Response:** Thanks for your suggestions. We have changed "impedes" to "reduces" to 105 soften the sentence. "the state of sub ice-shelf melting" refers to the basal melting state of the ice shelf, and we have revised it to "the state of sub-ice shelf melting" for enhanced clarity.

**16:** replace "inaccuracies" by "uncertainties"

**Response:** Thanks for your suggestions. We have revised the corresponding expression.

110 **17-18:** "we adopt a single ice sheet model (PISM) and investigate two different sub-ice shelf melt rate schemes during model spin-ups" - to me this implies the authors have performed experiments with two different sub-shelf melt schemes, but this is unclear in the experimental setup and results sections.

**Response:** Thanks for your suggestions. We have revised the relevant content in 115 Section 2 (Model and Methods) as follows:

*"During initialization procedure, to evaluate the specific role of oceanic conditions, we conducted two experiments using PISM: Experiment "S2" replicates the single simulation from LOW21 that used the best-fit parameter set (the one minimizing mismatch with observations), employing a thermodynamic parameterization (Eq. 2) to estimate sub-ice shelf melt rates. Experiment "S1" uses the same model configuration—including all parameters, stress balance approximation, resolution, topography, and atmospheric conditions—but replaces the basal melting scheme with observed basal melt rates derived from satellite altimetry (ICESat-1), radar (OIB and ALOS PALSAR), and model outputs (RACMO2), based on Eq. 1."*.

**21, 25:** "ice sheet regions" should be "ice-sheet regions", and "ice sheet mass changes" should be "ice-sheet mass changes". This should be modified throughout the paper (see lines 29)

**Response:** Thanks for your suggestions. We have made the corresponding changes — 4 in total across the manuscript.

**Introduction**

**27:** "discharge" should be "discharges"

**Response:** Thanks for your suggestions. We have revised the content.

**30-31:** "The exposure of ice shelves to warm seawater causes basal melting, combined with their near-flotation elevations, resulting in high susceptibility to oceanic forcing" - this sentence does not seem grammatically correct. Also, "vulnerability" seems more appropriate than "susceptibility"

**Response:** Thanks for your suggestion. We have revised the expression as follows: *"Ice shelves are highly vulnerable to oceanic forcing due to both basal melting from warm seawater and their near-flotation elevations (Bindschadler et al., 2013; Depoorter et al., 2013; Li et al., 2023)."*.

**34:** "grounding line retreat" should be "grounding-line retreat" here and elsewhere

**Response:** Thanks for your suggestions. We have made the corresponding changes — 4 in total across the manuscript.

**36-37:** I don't think Rignot et al (2008) is so explicit about MISI

**Response:** Thanks for your suggestions. We have revised the text and updated it with new references as follows: *"Particularly on retrograde bed slopes, such retreat may trigger Marine Ice Sheet Instability (MISI), a critical feedback mechanism often identified as a decisive factor in the collapse of the West Antarctica (Schoof, 2007; Hill*

*et al., 2024).".*

**38:** I think these two papers rather focus on MICI, not MISI (even if MISI is present too)

**Response:** Thanks for your suggestions. We have modified the content and updated the supporting references: *"This process may amplify Antarctic contribution to global sea-level rise by 0.5–0.8 m of sea level equivalent (SLE) this century (Ritz et al., 2015).".*

**39:** should be "ice-sheet model"; same in line 47, 50, 53

**Response:** Thanks for your suggestions. We have made the corresponding changes — 11 in total across the manuscript.

**40:** should be "parameterizations" (plural). Also, the authors could refer here to Favier et al (2019), who provide a review of sub-shelf melting parameterizations (Favier,L., Jourdain, N. C., Jenkins, A., Merino, N., Durand, G., Gagliardini, O., Gillet-Chaulet, F., and Mathiot, P.: Assessment of sub-shelf melting parameterisations using the ocean–ice-sheet coupled model NEMO(v3.6)–Elmer/Ice(v8.3), Geosci. Model Dev., 12, 2255–2283, https://doi.org/10.5194/gmd-12-2255-2019, 2019.).

**Response:** Thanks for your suggestions.

1. We have revised the "parameterizations".
2. We have added a citation to Favier et al. (2019), which reviews five parameterizations of sub-ice shelf melting—including three linear or non-linear thermal forcing schemes and two ice cavity models ("box" and "plume")—to the relevant sentence: *"Methods for ice-sheet models to represent sub-ice shelf melting include linear/non-linear and local/non-local dependency thermal forcing parameterizations (Martin et al., 2011; Favier et al., 2019; Lowry et al., 2021), ice-shelf cavity models developed from box or plume models (Lazeroms et al., 2018; PICO, Reese et al., 2018; Favier et al., 2019; PICOP, Pelle et al., 2019),".*

**44:** You should refer here specifically to initMIP-Antarctica. Also "ice sheet models exhibit significant divergence" - I guess you mean projections or ice-sheet model responses in experiments

**Response:** Thank you for your suggestion. We have revised it to *"The initMIP-Antarctica experiments revealed that ice-sheet model responses exhibit significant divergence due to variations in initial basal melt conditions. This uncertainty range accounted for 5 % to 125 % of total mass change in the initialization experiments (Seroussi et al., 2019, 2020).".*

**50:** I understand that this is an attempt to explain how this manuscript contributes beyond the results of initMIP-Antarctica. If so, please make this more explicit.

**Response:** Thank you for your suggestion. We have revised the relevant expression as

follows: *"Previous model intercomparison projects (e.g., initMIP-Antarctica) combined ice-sheet models with varying numerical complexities and initialization methods, making it difficult to attribute uncertainties to specific sources; our study isolates the impact of oceanic forcing by using a single model with identical initialization except for the basal melt scheme.".*

**56:** "the identical ice sheet model" seems to refer to CISM which is not the case. I suggest writing "a single ice-sheet model and initialization method"

**Response:** Thanks for your suggestions. Yes, "the identical ice-sheet model" refers to PISM as mentioned below. We have revised it to "a single ice-sheet model" for greater clarity.

**58:** I would rephrase question (2) as "How does this initial state affect long-term Antarctic ice sheet projections" - because I assume the basal melting scheme in the projections is the same as in LOW21 (see below).

**Response:** Thanks for your suggestions. We have revised the expression. We did indeed use the same basal melting scheme as LOW21 in the projections.

**60:** "Therefore, in this paper, we consider two different sub-ice shelf melt rate approaches" - it seems to me here you mean you will do experiments with both, but this is not clear after. Also, write "Parallel Ice Sheet Model, (PISM)" before "ice-sheet model"

**Response:** Thanks for your suggestions.

1. We have revised the relevant content in Section 2 (Model and Methods) as follows: *"During initialization procedure, to evaluate the specific role of oceanic conditions, we conducted two experiments using PISM: Experiment "S2" replicates the single simulation from LOW21 that used the best-fit parameter set (the one minimizing mismatch with observations), employing a thermodynamic parameterization (Eq. 2) to estimate sub-ice shelf melt rates. Experiment "S1" uses the same model configuration—including all parameters, stress balance approximation, resolution, topography, and atmospheric conditions—but replaces the basal melting scheme with observed basal melt rates derived from satellite altimetry (ICESat-1), radar (OIB and ALOS PALSAR), and model outputs (RACMO2), based on Eq. 1.".*

2. We have revised the expression to *"Therefore, in this paper, we consider two different sub-ice shelf melt rate approaches (Section 2) in the Parallel Ice Sheet Model (PISM) by first spinning-up and then projecting the AIS.".*

**61:** "the Antarctica ice sheet" should be "Antarctic ice sheet"

**Response:** Thanks for your suggestions. We have revised the content and have specified both the full name and the abbreviation of the Antarctic Ice Sheet in Line 56, and the "AIS" is used afterwards.

**Model and Methods**

**72:** LOW21 appears here for the first time. I understand that it refers to Lowry et al (2021). This should be stated clearly. More importantly, it is unclear to me if the authors have used the data from Lowry et al (2021) here or if they have redone the simulations using the same basal melting scheme as in that study. In this section and throughout the manuscript the authors refer to the results using the prescribed basal melting of Rignot et al (2013) (S1) as "our approach", or "our simulations" so it seems the simulations with parameterized melting scheme (S2) have not been done here. This is critical because it seems to me that in order to be sure the results shown are only due to the different sub-shelf melting and initialization the experiments need to be redone with exactly the same model version and configuration.

**Response:** Thanks for your suggestions.

1. We have revised the content to "LOW21 (Lowry et al., 2021)".

2. Prior to the initialization experiments using S1, we repeated the simulations based on the data and model configuration from Lowry et al. (2021), employing the TF-linear parameterization (S2). This procedure ensures that, apart from oceanic conditions, all other settings remain consistent with LOW21 when employing the observed basal melt rates (S1) (Fig. R1a). We are deeply grateful to Prof. Lowry for his patient guidance and support during this process, particularly in parameter setup and result validation, which enabled the successful replication of his experiments. Out of respect for Prof. Lowry's original work and to emphasize that our role was solely to replicate—not develop—the methodology, we attributed the configuration to LOW21 in the manuscript rather than highlighting our S2-based results.

3. During initialization, when using the S1, model configuration—including parameters, stress approximation, resolution, initial topography, and atmospheric conditions—is identical to LOW21. The only difference lies in the oceanic initial condition: our simulation uses observational sub-ice shelf melt rates, whereas LOW21 (S2) employed ocean temperature and salinity. In the projection experiments, both the model configuration and climatic forcing are the same (Fig. R1b).

4. Per your suggestions, we have clarified differences in the initialization and projection experiments. We have revised the relevant description as follows:

   *"During initialization procedure, to evaluate the specific role of oceanic conditions, we conducted two experiments using PISM: Experiment "S2" replicates the single simulation from LOW21 that used the best-fit parameter set (the one minimizing mismatch with observations), employing a thermodynamic parameterization (Eq. 2) to estimate sub-ice shelf melt rates. Experiment "S1" uses the same model configuration—including all parameters, stress balance approximation, resolution,*

*topography, and atmospheric conditions—but replaces the basal melting scheme with observed basal melt rates derived from satellite altimetry (ICESat-1), radar (OIB and ALOS PALSAR), and model outputs (RACMO2), based on Eq. 1.".*

*"We employed the same daily-resolution climate forcing as LOW21 (Lowry et al., 2021), derived from the CMIP5 IPSL-CM5A-MR RCP2.6/8.5 (Barthel et al., 2020; Payne et al., 2021; Nowicki et al., 2021) and the CMIP6 CNRM-CM6-1 SSP1-2.6/5-8.5 product (Nowicki et al., 2016; Kamworapan et al., 2021; Nowicki et al., 2021) spanning 2015–2100, to assess and compare Antarctica's contribution to global mean sea-level rise by 2100. To ensure that differences in projections originated solely from the model spin-up, the basal melting scheme was parameterized using the same linear thermodynamic framework for the ice shelf–ocean boundary layer as that employed in LOW21. This approach explicitly resolves heat and freshwater exchange processes at the ice–ocean interface, driven by oceanic forcing under different RCP/SSP scenarios from 2015 to 2100.".*

**73:** Figure 2 is referred to here before referring to Figure 1.

**Response:** Thank you for your suggestion and correction. We have changed "Fig. 2" to "Fig. 1".

**74:** "and other datasets" is vague, please be more specific

**Response:** Thank you for your suggestion. We have revised it to *"satellite altimetry (ICESat-1), radar (OIB and ALOS PALSAR data), and model outputs (RACMO2),".*

**80:** Maybe I am missing something but I don't fully understand how the basal temperatures are being used if the basal melting rates are imposed.

**Response:** Thanks for your suggestions.

1. In PISM, both S1 and S2 belong to the ocean model component, which supplies the ice dynamics component with sub-ice shelf basal temperature and mass flux during simulation. The sub-ice shelf basal temperature serves as a Dirichlet boundary condition for the energy equation within the ice dynamics system, while the basal mass flux acts as a source in the mass conservation equation. A positive flux indicates ice loss. S1 directly provides these two variables using observed basal melt rates and temperatures, whereas S2 indirectly supplies them through a TF-linear parameterization driven by ocean temperature and salinity.

2. To improve clarity, we have revised the content to *"The PISM ocean module provides the sub-ice shelf temperature and mass flux to the ice dynamics core via two different approaches. Sub-ice shelf temperature is applied as a Dirichlet boundary condition in the energy conservation code, while sub-ice shelf mass flux enters as a source in the mass conservation equation.".*

**104:** What does "evolutionary" mean here?

**Response:** Thanks for your suggestions.

1. The term "evolutionary" is ambiguous in this context. It refers to the third stage of the spin-up procedure, which is a 1,500-year model run employing the full model physics—including hybrid stress balance, calving law, and the application of sub-ice shelf melt rates to constrain ice dynamics.

2. The purpose of this stage is not to simulate a real-world evolution, but rather to allow the ice-sheet geometry—including thickness, velocity, and grounding line—to equilibrate dynamically and reach a steady state in the ice-sheet model under full model physics and prescribed climatic forcing, following the initial thermal equilibration.

3. To improve clarity, we have revised the expression to *"a 1,500-year model run incorporating full model physics"*.

**107:** Importantly: Please explain how sub-shelf melting is handled in the projections. I assume S2 is being used for both initializations but I did not see it written explicitly.

**Response:** Thanks for your suggestions.

1. The sub-ice shelf melt rates from Rignot et al. (2013) were used solely during the initialization in S1 to construct a new ice-sheet initial state under this oceanic condition, enabling comparison with S2 (LOW21) and analysis of resultant dynamic mechanism differences.

2. Furthermore, our study focuses on how variations in sub-ice shelf melt rates affect the initial ice-sheet state after spin-up and subsequently influence projected sea-level contributions. To maintain consistency with the LOW21 for comparative analysis in projection experiment, we used the same future oceanic forcing—specifically ocean temperature and salinity from CMIP5 and CMIP6 climate models—to project future ice-mass change.

3. To improve clarity, we have revised the expression to *"To ensure that differences in projections originated solely from the model spin-up, the basal melting scheme was parameterized using the same linear thermodynamic framework for the ice-shelf–ocean boundary layer as that employed in LOW21. This approach explicitly resolves heat and freshwater exchange processes at the ice–ocean interface, driven by oceanic forcing under different RCP/SSP scenarios from 2015 to 2100."*.

**107-108:** "high" and "low" are not scenarios but different grounding-line parameterizations using sub-grid interpolation of basal melting at grounding lines or not; this should be better explained. Also, was this also used in LOW21? This is also

critical to assess the differences

**Response:** Thanks for your suggestion.

1. Simulating retreat processes of marine-terminating glaciers in coarse-resolution grid models, the sub-grid scheme calculates one-sided derivatives of the surface slope around the grounding line and interpolates key physical variables based on spatial gradients across the interface between grounded and floating cells. Assign 0 to ice-free/floating cells, 1 to fully grounded cells, and 0–1 to partially grounded cells (includes grounding line). The formula for basal melt rate adjusted using this scheme is:

$$M_{b,adjusted} = \lambda M_{b,grounded} + (1 - \lambda)M_{b,shelf-base}$$

   $M_{b,grounded}$, $M_{b,shelf-base}$ denote the basal melt calculated for grounded ice grid cells and floating ice grid cells, respectively. $\lambda$ indicates the value (0-1) of the mask corresponding to the grid cell. This scheme is also used to adjust the basal friction in the transition zone. For more accurate expression, we have revised the term "sub-grid melt interpolation scheme" to "sub-grid grounding-line scheme" throughout the manuscript.

2. Yes, the terms "high" and "low" do not refer to climate scenarios, but to simulation results obtained by enabling or disabling the sub-grid scheme in PISM. For clarity, we have replaced the original terms "high scenario" and "low scenario" with the more descriptive expression "sub-grid scheme on (SGO) scenario" and "sub-grid scheme off (SGF) scenario". The "SGO scenario" activates this sub-grid melt interpolation, thereby accounting for basal melting in grid cells containing the grounding line. This results in higher overall mass loss because it accelerates grounding-line retreat within the coarse-resolution grid; without it, the retreat would not be simulated. By contrast, the "SGF scenario" omits the scheme and applies no basal melting to any grid cell that is not entirely floating. These neglects melting in partially floating cells, leading to lower total mass loss and thus representing a more conservative (lower melt) scenario.

3. LOW21 utilizes a statistical emulator for its projections. During initialization, model ensemble members were optimized using different parameter combinations related to ice flow and basal sliding. These parameters were evaluated both individually and collectively, with the optimal parameter set determined through validation against observation. Based on this initialized ensemble, the projection experiments utilized Gaussian process regression (GPR) to investigate the ice sheet's contribution to sea-level change.

4. In contrast to LOW21, our study did not employ a parameter ensemble optimization in spin-up, and the limited sample size precludes the GPR for projections statistical analysis. Therefore, we applied the "sub-grid" scheme to define projection ranges, adopting the optimal parameter set from LOW21 and selecting a climate forcing identical to one of their ensemble members. S1-based projection results are directly compared against the corresponding ensemble results from LOW21, which include

the S2-based projection (Figs. 8, 9).

5. We have supplemented this section accordingly and modified the relevant description in the revision as follows:

*"Further, based on the initialized model state and the optimal parameter set from S1, we conduct projection experiments from 2015 by turning on or off the sub-grid grounding-line scheme in PISM. The "sub-grid scheme on (SGO) scenario" incorporated sub-grid melt interpolation near grounding lines, accelerating grounding-line retreat in our coarse-resolution model, while the "sub-grid scheme off (SGF) scenario" ignored melt in partially floating cells, yielding more conservative mass loss estimates (Albrecht et al., 2011; Golledge et al., 2015; Nowicki et al., 2020).".*

**Figure 1:** Does "our simulations" mean S1? Also, the temperature field of Chambers et al (2021) should be referred to in the text.

**Response:** Thanks for your suggestion. Yes, we use Eq. 1 to simulate the basal mass flux of the ice shelf. We have revised it to *"Where directly observed basal melt rates (Rignot et al., 2013; Fig. 1) and ice-shelf basal temperature (Chambers et al., 2021; Fig. 1) are used, the sub-ice shelf mass flux is computed directly using Eq. 1.".*

**Figure 2:** Again, authors say "our study and LOW21" so it is unclear if they have performed simulations using S2. Also, the discussion in lines 205-208 below on the fact Rignot et al (2013)'s sub-shelf melting rates are higher than those used in S2/LOW21 should appear here, when the differences are shown (see comment below)

**Response:** Thanks for your suggestion.

1. For clarity and to accurately reflect that our simulations utilized the S2 approach, we have revised the figure caption and description as follows: *"Figure 2: Comparison of sub-ice shelf melt rates between S1 and S2. (a) Sub-ice shelf melt rates derived from the TF-linear parameterization (S2). (b) Difference in basal melt rates used in S1 and S2, with three black boxes highlighting regions of interest: (c) Thwaites Basin, (d) Wilkes Land, and (e) George V Land Terre Adelie.".*

2. Based on your feedback, we have moved the discussion regarding the fact "that Rignot et al. (2013)'s sub-shelf melt rates are higher than those used in S2/LOW21" from Section 3.4 to Section 3.2: *"In the Thwaites Basin, the observed sub-ice shelf melt rates used in S1 (reaching 17.7 m y$^{-1}$ beneath Thwaites ice shelf, Fig. 1) exceed S2's parameterized values by approximately 5 m y$^{-1}$ (Fig. 2). This higher melt rate weakens the ice-shelf buttressing effect and accelerates the grounded ice flow, with a corresponding 74 m y$^{-1}$ RMSE difference from S2 (Fig. 5f).".*

**Model Initialization Results**

**114:** Again, this sounds as if you have not done the simulations with S2.

**Response:** Thank you for your suggestion.

410 1. We have revised the description to clarify that the simulation was conducted using method S2 in Section 2 as follows: *"Experiment "S2" replicates the single simulation from LOW21 that used the best-fit parameter set (the one minimizing mismatch with observations), employing a thermodynamic parameterization (Eq. 2) to estimate sub-ice shelf melt rates. Experiment "S1" uses the same model*

415 *configuration—including all parameters, stress balance approximation, resolution, topography, and atmospheric conditions—but replaces the basal melting scheme with observed basal melt rates derived from satellite altimetry (ICESat-1), radar (OIB and ALOS PALSAR), and model outputs (RACMO2), based on Eq. 1.".*

   2. To improve clarity, we have revised this sentence to: *"We validated the simulated*

420 *ice thickness and ice surface velocity results from both S1 and S2 using observational datasets (BedMachine v.3; MEaSUREs Phase-Based Antarctica Ice Velocity Map v.1).".*

**115:** Start a new sentence to describe the RMSE differences (between S1 and S2/LOW21?)

425 **Response:** Thank you for your suggestion. We have revised it to *"The difference in root mean square error (RMSE) for two experiments, derived from comparison against observation, is 2 m for ice thickness and 3 m y$^{-1}$ for surface velocity.".*

**118:** "consistent" sounds weak - I guess you mean very similar between S1 and S2

**Response:** Thanks for your suggestion. We have revised it to *"The comparison shows*

430 *that the mass distribution and ice flow dynamics of S1 closely match those of S2.".*

**118-121:** Figure 7 is referred to here (before Figures 3-6). There is no need for this now, since you actually only describe these results later on.

**Response:** Thanks for your suggestion. We have removed the reference to Fig. 7 and changed "selected a transect (Fig. 7)" to "selected a representative transect".

435 **122:** Antarctica should be Antarctic and ice sheet should be ice-sheet

**Response:** Thanks for your suggestions. We have revised the content.

**129:** I think there is no need for a new paragraph here. Also, here and elsewhere, I would rather use the notation S1 and S2 rather than "our" and "LOW21"

**Response:** Thanks for your suggestions.

440 1. We have merged this paragraph with the previous content into one paragraph.

   2. Based on your suggestion, we have revised Section 2 to clarify that two distinct methods (labeled "S1" and "S2") are used to calculate basal fluxes, replacing the original terms "our study" and "LOW21": *"Experiment "S2" replicates the single simulation from LOW21 that used the best-fit parameter set (the one minimizing*

*mismatch with observations), employing a thermodynamic parameterization (Eq. 2) to estimate sub-ice shelf melt rates. Experiment "S1" uses the same model configuration—including all parameters, stress balance approximation, resolution, topography, and atmospheric conditions—but replaces the basal melting scheme with observed basal melt rates derived from satellite altimetry (ICESat-1), radar (OIB and ALOS PALSAR), and model outputs (RACMO2), based on Eq. 1.".*

3. These revisions have been applied consistently throughout the manuscript, including figures and tables.

**145:** replace "led" by "leads" (to have everything in the same verbal tense, present)

**Response:** Thanks for your suggestions. We have revised this expression.

**149:** What complex ice-dynamic feedbacks?

**Response:** Thanks for your suggestions.

1. The term "complex ice-dynamic feedbacks" primarily refers to the positive feedback triggered by increased basal melt rates—the "melt–buttressing reduction–flow acceleration–grounding-line retreat and ice thinning" process. Specifically, as mentioned earlier regarding the Thwaites Basin, the higher basal melt rates applied in our simulation—compared to those used in LOW21— weakened ice-shelf buttressing, accelerated ice flow, and led to grounding-line retreat and localized ice thinning.

*2.* To improve clarity, we have revised the content to describe the process more directly as follows: *"This leads to around 40 m more ice thinning near the grounding line and an approximately 30 km more grounding-line retreat (Fig. 7), compared to the case in LOW21, while most upstream areas exhibit positive thickness anomalies (mean 49.5 m), indicating a coupled response within the ice-sheet system.".*

**152:** Totton should be Totten

**Response:** Thanks for your suggestions. We have revised the content.

**154-156:** I am not sure about this mechanism. Thinning downstream causes thickening upstream? Can this be explained further via a reference or additional results showing the stronger lateral resistance invoked?

**Response:** Thanks for your suggestions. We have added references to explain this process through lateral resistance. The revised content is as follows: *"The faster flow of Totten Glacier strengthens lateral resistance along its boundaries with adjacent glaciers, subsequently reducing ice discharge into the Voyeykov and Moscow Ice Shelves (Gagliardini et al., 2010; Van Der Veen et al., 2017). This dynamic response is consistent with the simulated mean thickness anomaly of +39.2 m across these regions,*

*indicating ice thickening resulting from slower flow.".*

**156:** Replace "decreased" by "decreases"

**Response:** Thanks for your suggestions. We have revised the content.

**174:** replace "reduced" by "are reduced"

**Response:** Thanks for your suggestions. We have revised the content.

**178-184:** To what extent are these processes represented in the ice-sheet model? For example, sedimentary basins? Also, replace "WAIS" by "the WAIS" and "EAIS" by "the EAIS"

**Response:** Thanks for your suggestions.

1. PISM employs a simplified approach, representing subglacial hydrology through a parameterized till reservoir system in which meltwater is stored within a saturated till layer of limited thickness. Local basal melting saturates till water content, meltwater exceeding maximum storage thickness drains directly into subglacial hydrologic networks.

2. While the model focuses primarily on large-scale ice dynamical processes, the geometry of subglacial sedimentary basins is implicitly captured through initial topography and ice thickness. Their thermal and hydrological influence is indirectly reflected in the simulated enthalpy field (governing ice temperature) and water content distribution. Explicit representation of sediment–hydrology interactions would require coupling with more physically detailed subglacial hydrological models (e.g., Wright et al., 2012).

3. We have made the corresponding changes (5 for "WAIS" and 2 for "EAIS") across the manuscript.

**192:** should be "ice-sheet destabilization"

**Response:** Thanks for your suggestions. We have revised the content.

**195:** The sentence starting with "This positive feedback…" seems grammatically incorrect

**Response:** Thanks for your suggestions. We have revised it to *"In this positive feedback process, termed the basal thermal-hydrological feedback, elevated basal water content persistently reduces resistance, thereby facilitating ice sliding and ultimately leading to ice thinning (Fowler et al., 2001; Clarke, 2005; van Pelt & Oleremans, 2012; Zhao et al., 2025).".*

**205:** Should be "the" WAIS and "the susceptibility"

**Response:** Thanks for your suggestions. We have revised the content.

**206-208:** It seems to me this discussion on the fact Rignot et al (2013)'s sub-shelf melting rates are higher than those used in S2/LOW21 should appear earlier, at the end of Section 2 when the differences are shown

**Response:** Thanks for your suggestions. We have revised the fact to Section 3.2 (Differences in Marine Ice-Sheet Regions): *"In the Thwaites Basin, the observed sub-ice shelf melt rates used in S1 simulation (reaching 17.7 m y$^{-1}$ beneath Thwaites ice shelf, Fig. 1) exceed S2's parameterized values by approximately 5 m y$^{-1}$ (Fig. 2). This higher melt rate weakens the ice-shelf buttressing effect and accelerates the grounded ice flow, with a corresponding 74 m y$^{-1}$ RMSE difference from S2 simulation (Fig. 5f)."*.

**212:** should be "ice-shelf buttressing"

**Response:** Thanks for your suggestions. We have made the corresponding changes—3 in total across the manuscript.

**Figure 3 and table 1:** When are these fields exactly taken? At the end of the initialization procedure? Do they correspond to 2015?

**Response:** Thanks for your suggestions. These fields correspond to the ice-sheet state at the end of the initialization (spin-up) process, representing its condition around 2015. To improve clarity, we have revised the description of Fig. 3 and Table 1 as follows: *"Figure 3 Comparing simulated initial state with observations. Modeled ice thickness (m) and surface ice velocity (m yr$^{-1}$) at the end of spin-up. The left column shows ice thickness and ice surface velocity results from S1, alongside their difference from observation (Morlighem et al., 2019; Mouginot et al., 2019). The right column shows the corresponding results from S2, sharing common color bars with S1."*.

*"Table 1 Ice volume above flotation (m SLE) in three marine ice-sheet basins after spin-up, simulated at 16 km resolution."*.

**Figure 4:** Are you using the same color bar and levels for ice thickness and velocity? This strongly limits visualization of the ice-thickness anomalies so I would suggest to use different color bars and levels for each field

**Response:** Thanks for your suggestions. It is possible that the questions you raised appear to be associated with Fig. 3, which was originally designed to compare the reliability of simulated results between two methods. The anomalies in ice thickness and surface velocity are illustrated in Fig. 5. To improve clarity, we have supplemented the description of Fig. 3 as follows: *"The left column shows ice thickness and ice surface velocity results from S1, alongside their difference from observation (Morlighem et al., 2019; Mouginot et al., 2019). The right column shows the corresponding results from S2, sharing common color bars with S1."*.

**Figure 7:** There is no discussion as to how isostasy is treated. Is the bedrock elevation identical at this stage for S1 and S2 or are you just showing the original data?

**Response:** Thanks for your suggestions.

1. All fields shown in Fig. 7 represent the ice-sheet state at the end of the spin-up process.

2. For consistency with LOW21, which did not use earth deformation options (i.e., no viscoelastic bedrock processes were considered), our simulations also did not account for glacial isostatic adjustment. Therefore, the bedrock elevation remains identical under both the S1 and S2 methods, so it is not distinguished separately in Fig. 7. In reality, the response of the ice-sheet bedrock to ice load changes occurs on millennial timescales or longer. To properly account for relevant parameters such as mantle viscosity and lithospheric flexural rigidity, a paleo-climate spin-up covering two glacial cycles—rather than the constant-climate spin-up used in S1 and S2—would be required in PISM.

3. We have revised the title of Fig. 7 as follows: *"Comparison between S1 and S2 along the Thwaites Basin transect after spin-up."*.

**Model Projection Results**

A general comment: Please describe how the thermal forcing is applied in these simulations (which basal melting scheme).

**Response:** Thanks for your suggestions.

1. To investigate whether differences in projections originate solely from the different initial states of S1 and S2 (LOW21), we maintained a single variable by using the same basal melting scheme (linear thermodynamic forcing parameterization) in our projections as in LOW21 (Fig. R1b, P.3). Unlike the S2 method used in initialization—which applies constant boundary conditions—the projection employs time-varying climate forcing. To distinguish between these two types of data, we use the term "condition" for constant inputs and "forcing" for time-dependent inputs.

2. We have supplemented the description of the basal melting scheme used in Section 2 (Model and Methods): *"To ensure that differences in projections originated solely from the model spin-up, the basal melting scheme was parameterized using the same linear thermodynamic framework for the ice shelf–ocean boundary layer as that employed in LOW21. This approach explicitly resolves heat and freshwater exchange processes at the ice–ocean interface, driven by oceanic forcing under different RCP/SSP scenarios from 2015 to 2100."*.

**231-233:** "Prognostic simulations from 2015 to 2100 revealed divergent ice mass changes compared to LOW21" - where is this shown? If it is just in Figure 8 this suggests the simulations for S2 have not been done again, is that the case? If so I think it would be very difficult to be able to guarantee that the model versions are identical (given that LOW21 was published four years ago) and the only difference is the initialization unless the simulations.

**Response:** Thanks for your suggestions.

1. Figure 8(a) depicts the difference in projected ice thickness distribution between S1 and S2 (the optimal-parameter simulation in LOW21 ensemble). The values represent the mean anomaly for the year 2050, derived under identical RCP scenarios. Furthermore, as suggested, We have added the sea-level time series from the S2-based projection (using the same optimal parameter set and forcing) to Figs. 8(c) and 9, where they are represented by dashed lines.

2. Our study focuses on how variations in sub-ice shelf melt rates affect the initial ice-sheet state after spin-up and subsequently influence projected sea-level contributions. We directly utilized the projection results provided by Prof. Lowry, which were generated using the same optimal parameter set and identical climate forcing as applied in our S1-based projections. This was feasible because the structure of the projection script is identical to the final step of initialization, allowing for a direct substitution of the climate forcing.

3. To ensure consistency of results, we used the same PISM version as in LOW21, which was verified through detailed discussions with Prof. Lowry regarding technical implementation details. Our team began using PISM around 2020, originally adopting an early version of PISM. Due to persistent hardware limitations, we were unable to update to newer versions in subsequent work.

4. We have revised the figures supporting this conclusion, and the supplemented sentence is as follows: *"Prognostic simulations (2015–2100) show that the divergent initial ice-sheet states of S1 and S2 lead to markedly different sea-level contributions across the AIS, even under identical climatic forcing and basal melt scheme (Figs. 8, 9). Specifically, S1-based projections of a 0.20–0.52 m SLE total AIS contribution exceed the 0–0.32 m SLE range of the LOW21 ensemble (which includes S2-based projections) by roughly 0.18 m SLE, representing a ~57% increase (Fig. 8).".*

**232:** replace "ranging" by "range"

**Response:** Thanks for your suggestions. We have revised the content.

**238-239:** what does "relative to the hysteretic response of ice-sheet dynamics to climate forcing" mean here? Please reformulate to clarify.

**Response:** Thanks for your suggestions.

1. This sentence explains the reason for the overlapping projections across scenarios before 2075. This is because the ice sheet's hysteretic response means that its full reaction to new climatic forcings takes considerable time to appear. Therefore, changes during this period primarily reflect the ice sheet's response to past historical forcing, rather than to divergent future emission scenarios.

2. In response to your comment, we have revised the statement as follows: *"This is consistent with the hysteretic response of ice-sheet dynamics, meaning that the ice sheet's state in the near-term (2015-2075) is largely determined by historical forcing, masking the influence of divergent future scenarios (Garbe et al., 2020).".*

**242:** 0.36 m is in the mean value, this should be stated

**Response:** Thanks for your suggestion. We have revised it to *"By 2100, the mean AIS contributions to sea-level rise under SSP5-8.5 reach 0.36 m SLE,"*.

**261-268:** The discussion starting with "This significant mass loss is propelled… "describes what we think has been the main mechanism driving Antarctic ice mass loss in the past decades, but it is discussed here as if it were also the relevant mechanism leading to ice-loss in the present simulations. There is however no support from figures to demonstrate that this indeed the case, so this is therefore speculative.

**Response:** Thanks for your suggestion. We agree with the reviewer that the mechanism should be discussed as a plausible interpretation rather than a confirmed finding from our simulations. We have revised the text accordingly to frame it as a consistent, literature-based inference: *"A mechanism similar to that observed in recent decades may be responsible for the projected mass loss. Specifically, anthropogenic warming could alter shelf-break wind patterns over the Amundsen Sea (AS) and Bellingshausen Sea (BS) Embayment (Holland et al., 2019; Noble et al., 2020), potentially facilitating greater intrusion of warm water and intensifying ice melting beneath ice shelves (Dinniman et al., 2016; Noble et al., 2020; Li et al., 2023). This would lead to reduced ice-shelf buttressing and accelerated ice discharge. The dominance of the AS and BS sectors in our projections (~55% of WAIS loss, Table 2) is consistent with the operation of such a mechanism."*.

**271-272:** Similarly, the enhanced moisture transport is not illustrated so this remains speculative.

**Response:** Thanks for your suggestion. As suggested, we have revised the discussion on the WIO sector as follows: *"The net mass gain in the West Indian Ocean (WIO) sector shown in our simulations may be linked to enhanced moisture transport from the Southern Ocean, a mechanism consistent with observational trends (Boening et al., 2012) that would promote increased surface accumulation."*.

**279-280:** The same applies to the sentence "This transient pattern is tied to the intensification of polar westerly winds"

**Response:** Thanks for your suggestion. As suggested, we have revised this paragraph to: *"A possible interpretation for this transient pattern involves the intensification of polar westerly winds. According to this mechanism, which is consistent with observational findings (Goodwin et al., 2016), enhanced snowfall in the northern AP may partially offset warming-induced ice discharge, thereby generating a negative feedback that suppresses AIS mass loss."*.

**292:** "the" WAIS

**Response:** Thanks for your suggestions. We have made the corresponding changes—7 in total across the manuscript.

**296:** Please introduce the emulator the first time Edwards et al (2021) is mentioned to give a little more context.

**Response:** Thanks for your suggestions. We have revised it to *"The Coupled Model Intercomparison Project Phase 6 models (CMIP6, Edwards et al., 2021) employed Gaussian process emulators—statistical approximations built upon ice-sheet simulations for ISMIP6 (Nowicki et al., 2016, 2020) and GlacierMIP Phase 2 (Hock et al., 2019)—to generate sea-level projections. While their ensemble projections suggest that the WAIS contributions range from -0.04 to 0.11 m SLE, our study predicts a significantly higher contribution of 0.20–0.47 m SLE."*.

**309:** I suggest writing "demonstrating its heightened vulnerability to ocean-induced melt rates at the initialization"

**Response:** Thanks for your suggestions. We have revised this expression.

**315, 318, 329, 331:** should be "ice-sheet"

**Response:** Thanks for your suggestions. We have revised these contents across the manuscript.

**321:** Indeed, the AIS might not be in steady state for the present day, and if projections are very sensitive to initial conditions one has to think what the most realistic state would be. For instance, reaching that initial state (previous to projections) via long, transient spin ups involving paleo-evolution could be critical because of the different dynamical state achieved in this way. It would be nice if the authors could discuss how they think a better initial state can be achieved.

**Response:** Thanks for your suggestions.

1. To achieve a better initial state in ice-sheet simulations, it is crucial to minimize uncertainties arising from differences in model parameterization and initialization methods. Results from the ISMIP6-Antarctica project indicate that ice-sheet model-related factors dominate these uncertainties. Therefore, improvements in model development and extensive sampling of parameter space are essential to better represent key physical processes.

2. In response to your feedback, we have expanded our discussion on the sources of uncertainty related to both ice-sheet models and observational data. Building on existing literature, we also propose directions for improving future simulations. The modified and supplemented paragraphs are as follows:

   *"Compared to other prior studies, our sea level projections differ due to variations in ice-sheet model configurations, including model resolution, ice dynamics (particularly stress balance schemes), represented physical processes (calving, hydrology, or bedrock uplift), and initialization methods (data assimilation or spin-up) (Seroussi et al., 2019; Levermann et al., 2020; Klose et al., 2024). Of these factors, the parameterization of ice melt dynamics contributes most significantly to*

*the uncertainty in sea-level estimates, surpassing uncertainties arising from differences in climate model forcing, initialization methods, and the selected physical processes. This implies that ice-model-related uncertainties dominate throughout the simulation period (Seroussi et al., 2019, 2023). Therefore, continual model improvement, further exploration of the broader parameter space covered by initial state ensembles, and its extended sampling are essential to reduce uncertainties in future projections of dynamic mass loss from the AIS (Favier et al., 2019; Coulon et al., 2024; Klose et al., 2024).".*

*"Notably, the present-day AIS may not have been in a steady-state during the observational period (Martin et al., 2011). This inference, while primarily based on discrepancies between model simulations and observations, may also be influenced by uncertainties inherent in the validation datasets. For example, the BedMachine v3 dataset relies on approximate calculations in regions such as ice-free land, ocean bathymetry, and cavities under ice shelves, potentially introducing spatial biases in thickness estimates (Morlighem et al., 2019). Similarly, the MEaSUREs velocity map inevitably contains errors in flow direction derived from phase data and speckle tracking during SAR data processing (Mouginot et al., 2019). Thus, the apparent model–data mismatch not only demonstrates the non-steady-state of AIS but also reflects the challenge of validating model simulations against modern records that contain their own uncertainties and potential biases. This underscores the need for obtaining more, highly accurate, and extensive observations for verification and validation is crucial to better constrain ice-sheet models and improve the reliability of future sea-level estimation (Seroussi et al., 2020; Seroussi et al., 2023).".*

**Table 2:** I suppose the numbers in brackets correspond to "high" and "low" projections and the individual numbers are averages but this should be stated in the table caption. Also, India should be Indian and Antarctica should be Antarctic.

**Response:** Thanks for your suggestions.

1. We have revised the content regarding the "Indian" and "Antarctic".
2. The confidence intervals provided in Table 2 summarize simulation results obtained by either enabling or disabling the sub-grid grounding-line scheme in the prediction process. When using the sub-grid scheme, the model applies weighted adjustments to ice mass changes in the ice sheet–shelf transition zone. This results in an "SGO scenario", which defines the upper bound of the confidence interval for the sea-level contribution in Table 2. Conversely, when this scheme is disabled, the model neglects these ice mass changes, yielding the "SGF scenario" that defines the lower bound of the interval. This methodology follows the approach of Golledge et al. (2015).
3. In response to your feedback, we have supplemented the description of Table 2: *"Sea level contribution (m SLE) of Antarctic Ice Sheet Basins by 2100. The*

*confidence intervals represent the range of sea-level contribution from the "SGO scenario to the "SGF scenario" simulation across different RCP/SSP scenarios; the single value denotes the mean value of this range.".*

**Figure 8:** I would have expected to see the same scenarios carried out with the alternative initial conditions.

**Response:** Thank you for your suggestion.

1. Figure 8 (a) shows the spatial differences in simulated mean ice thickness between S1 and S2 (a set of ensemble projections from the LOW21) under RCP 2.6 and 8.5 scenarios, while (b) displays the spatial differences of the ensemble mean for the year 2100.

2. We have supplemented Fig. 8 (c) with the corresponding time series from the S2-based projection. These time series—a set of projections from LOW21 obtained under the IPSL-CM5A-MR RCP2.6 and RCP8.5 scenarios, using the same parameter configuration as the S1-based projection—are depicted as dashed lines. Since the experimental configuration and model options are identical between the initialization and projection simulations, we only repeated the initialization to verify the correct setup, and were unable to replicate the full set of projection results. We are grateful to Prof. Lowry for providing these scenario-specific projection results.

3. To improve clarity, we have revised the description in Fig. 8:
   *"Spatial differences in the projected mean ice thickness between the multi-scenario (RCP 2.6 and RCP 8.5) ensemble means of S1 and S2 in 2050 (a) and 2100 (b). (c) Predicted sea level rise for "SGO scenario" to "SGF scenario" simulations under four scenarios (color shading) and mean values (color lines). The dashed lines represent projections from the S2 initial state—a set of results from the LOW21 ensemble: red for RCP8.5, blue for RCP2.6.".*

[Figure]

**Figure 9:** Similar to my comment above, I would have expected to see similar results for the alternative initial conditions. India should be Indian.

**Response:** Thank you for your suggestion.

1. We have revised the content regarding the "Indian".

2. To facilitate comparison, we have added to the Fig.9 the S2-based time series (a set of projections from LOW21) of the AIS contribution to sea-level rise under RCP 2.6 and RCP 8.5 for individual drainage basins. These projections, which use the same parameters and climate forcing as S1-based projection, are depicted as dashed lines corresponding to each scenario.

3. Corresponding descriptions have been added in the text: *"The solid and dashed lines represent projections from the S1 and S2 initial states, respectively, under different climate scenarios, with the S2 projections being part of the LOW21 ensemble (red for RCP8.5, blue for RCP2.6)."*.

[Figure]

**Response:** Thanks for your suggestions. We have plotted a time series figure that displays the volume above flotation (VAF) during both the historical phase in spin-up and projection results under the RCP/SSP scenarios simulated using method S1.

[Figure]

**Figure R2: Ice Volume above Flotation in Spin-up and Projection.** Time series show S1-based historical period in spin-up (black), and projections under RCP/SSP scenarios (light blue, red, dark blue, orange) and constant-climate control projection (grey).

**Response:** Thanks for your suggestions. We have revised this content.

**References:**

Favier, L., Jourdain, N. C., Jenkins, A., Merino, N., Durand, G., Gagliardini, O., Gillet-Chaulet, F., & Mathiot, P. (2019). Assessment of sub-shelf melting parameterisations using the ocean–ice-sheet coupled model NEMO(v3.6)–Elmer/Ice(v8.3). *Geoscientific Model Development*, *12*(6), 2255-2283. https://doi.org/10.5194/gmd-12-2255-2019

Gagliardini, O., Durand, G., Zwinger, T., Hindmarsh, R. C. A., & Le Meur, E. (2010). Coupling of ice-shelf melting and buttressing is a key process in ice-sheets dynamics. *Geophysical Research Letters*, *37*(14). https://doi.org/10.1029/2010gl043334

Hill, E. A., Gudmundsson, G. H., & Chandler, D. M. (2024). Ocean warming as a trigger for irreversible retreat of the Antarctic ice sheet. *Nature Climate Change*, *14*(11), 1165-1171. https://doi.org/10.1038/s41558-024-02134-8

Hock, R., Bliss, A., Marzeion, B. E. N., Giesen, R. H., Hirabayashi, Y., Huss, M., RadiĆ, V., & Slangen, A. B. A. (2019). GlacierMIP – A model intercomparison of global-scale glacier mass-balance models and projections. *Journal of Glaciology*, *65*(251), 453-467. https://doi.org/10.1017/jog.2019.22

Leguy, G. R., Asay-Davis, X. S., & Lipscomb, W. H. (2014). Parameterization of basal friction near grounding lines in a one-dimensional ice sheet model. *The Cryosphere*, *8*(4), 1239-1259. https://doi.org/10.5194/tc-8-1239-2014

Van Der Veen, C. J., Stearns, L. A., Johnson, J., & Csatho, B. (2017). Flow dynamics of Byrd Glacier, East Antarctica. *Journal of Glaciology*, *60*(224), 1053-1064. https://doi.org/10.3189/2014JoG14J052